# TACO-NET: TOPOLOGICAL SIGNATURES TRIUMPH IN 3D OBJECT CLASSIFICATION

## ABSTRACT

3D object classification is a crucial problem due to its significant practical relevance in many fields, including computer vision, robotics, and autonomous driving. Although deep learning methods applied to point clouds sampled on CAD models of objects and/or captured by LiDAR or RGBD cameras have achieved remarkable success in recent years, achieving high classification accuracy remains a challenging problem due to the unordered point clouds and their irregularity and noise. To this end, we propose a novel state-of-the-art (SOTA) 3D object classification technique that combines topological data analysis with various image filtration techniques to classify objects when they are represented using point clouds. We transform every point cloud into a voxelized binary 3D image to extract distinguishing topological features. Next, we train a lightweight one-dimensional Convolutional Neural Network (1D CNN) using the extracted feature set from the training dataset. Our framework, `TACO-Net`, sets a new state-of-the-art by achieving $99.05\%$ and $99.52\%$ accuracy on the widely used synthetic benchmarks ModelNet40 and ModelNet10, and further demonstrates its robustness on the large-scale real-world OmniObject3D dataset. `TACO-Net` also achieved one of the highest accuracies among all the from-scratch supervised learning techniques when tested on the hardest variant of the ScanObjectNN dataset. When tested with ten different kinds of corrupted ModelNet40 inputs, the proposed `TACO-Net` demonstrates strong resiliency overall.

## 1 INTRODUCTION

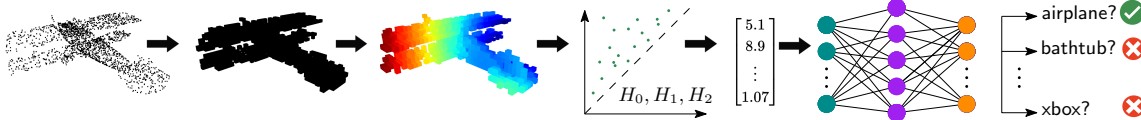

Figure 1: An airplane point cloud (from ModelNet40) is converted into a 3D binary image, which is then transformed into a set of 3D grayscale images (just one is shown) using different filtration techniques. Every grayscale image admits a separate cubical persistence. A feature vector of length 36 is obtained for every persistence. The vectors are then concatenated to form the final feature vector for the plane. The feature vectors are finally trained using a 1D CNN for object classification.

Semantic object recognition is one of the most fundamental capabilities that modern-day autonomous systems, such as robots and cars, demand to operate in dynamic real-world environments. Given an input point cloud, the objective is to classify the input into one of the known categories Maturana & Scherer (2015); Qi et al. (2017a); Wu et al. (2015). Such object classification is critical in numerous real-world applications, including autonomous driving, robotics, augmented reality, and 3D scene understanding. In recent years, deep learning techniques have achieved remarkable success in the 3D object classification task by using various modes of inputs such as voxels Maturana & Scherer (2015), multi-views Kanezaki et al. (2019), and raw point clouds Qi et al. (2017a), to name a few.

Unlike 2D images, point clouds provide rich geometric and spatial information in three dimensions, enabling more precise object recognition in complex environments Sarker et al. (2024). Further, unlike 2D images, point clouds enable a mobile robot, for example, to recognize objects in various ambient light and weather conditions efficiently. However, point cloud-based object classification remains challenging due to the unstructured and sparse nature of point cloud data, which lacks a regular grid structure and often suffers from noise, occlusion, and varying point densities Ben-Shabat et al. (2018); Sun et al. (2022); Uy et al. (2019). Furthermore, the permutation invariance of points and the need to capture both local and global geometric features add layers of complexity to model design Zhao et al. (2022). These factors demand innovative methodologies that effectively learn from unordered and irregular 3D geometric data, driving continued research and development in this field.

Most of the existing approaches use a deep machine learning framework where the input is either a set of pictures of the object, a raw set of points, or volumetric shape of the object Su et al. (2015); Feng et al. (2018); Kanezaki et al. (2019); Qi et al. (2017a;b); Qian et al. (2022); Maturana & Scherer (2015). Unlike these, we voxelize any $n$-point 3D object cloud once to compute persistent-homology signatures via cubical persistence, reframing topological descriptors as a universal substrate for deep representation learning. From these fixed, pre-computed signatures, our lightweight 1D CNN learns a highly structured, class-disentangled latent space for 3D shape recognition, yielding strong separability without requiring gradients to flow through the front-end topological data analysis (TDA) computations. The 1D CNN is then trained end-to-end on the resulting topological feature vectors for the final classification task. See Fig. 1 for a system-level overview of our approach. Although TDA has a direct connection with shapes, surprisingly, TDA has not been successfully used for large-scale 3D object classification. Similar to the challenges associated with designing deep learning models using existing network layers, finding an effective TDA pipeline presents a challenge. First, we test the proposed **t**opological data **a**nalysis-based obje**c**t classification framework, named `TACO-Net`, on ModelNet40 and ModelNet10 datasets. Our experiments show that we achieve SOTA accuracy for both these datasets. Furthermore, we have chosen ten common corruptions to test the robustness of `TACO-Net`. The corrupted dataset is tested on the model trained with the uncorrupted ModelNet40 dataset.

Results show that the proposed `TACO-Net` yields high accuracy for all but one corruption type at a low level while achieving moderate accuracy with highly corrupted test data. Further, when tested on a real-world dataset, namely OmniObject3D Wu et al. (2023), `TACO-Net` again achieved the highest accuracy. We tested `TACO-Net` on the six popular variants of ScanObjectNN. It achieved the second-highest accuracy of 91.87% among all the existing supervised learning techniques for the hardest variant PB_T50_RS. For the OBJ_BG and OBJ_ONLY variants, it achieved the SOTA accuracies of 95.08% and 94.01%, respectively. To show the generalizability of `TACO-Net`, we tested it on two 3D medical object datasets, namely VesselMNIST3D Yang et al. (2020) and AdrenalMNIST3D Yang et al. (2023), where `TACO-Net` surpassed the highest accuracies and F1-scores of numerous existing techniques such as PointNet Qi et al. (2017a), PointNet++ Qi et al. (2017b), and DGCNN Wang et al. (2019), among others. The main contributions of our paper are as follows.

- To our knowledge, this is the first 3D classification framework that learns a discriminative latent space over fixed cubical persistent-homology signatures using a lightweight 1D CNN that effectively internalizes and amplifies non-learned topological descriptors.
- Our proposed `TACO-Net` framework achieves state-of-the-art accuracy on ModelNet10/40, and two challenging real-world benchmarks - OBJ_BG and OBJ_ONLY of ScanObjectNN - demonstrating that it effectively learns class-disentangled structure over topological inputs, and while performing competitively on the hardest variant of ScanObjectNN.
- `TACO-Net` demonstrates strong robustness to severe point-cloud corruptions and generalizes across multiple real-world benchmarks and their variants, a scale of cross-dataset validation rarely reported in 3D object-classification literature.

**Related Works.** Three main types of approaches are prevalent in the 3D object classification literature: voxel-based, multi-view imaging-based, and raw point-based Sarker et al. (2024). Many hybrid methods

combine one or more of the above-mentioned techniques. In voxel-based methods, features of the volumetric representation of input point clouds are learned and classified Maturana & Scherer (2015). One of the earliest approaches in this direction is 3D Shapenets Wu et al. (2015). Although the objects to be classified are in 3D, taking 2D pictures of them from various angles and classifying those 2D pictures instead has gained attention through MVCNN by Su et al. Su et al. (2015), where they used 80 pictures of each 3D object. GVCNN Feng et al. (2018) improved upon MVCNN by using only 8 images. MHBN Yu et al. (2018), on the other hand, used only 6 views of the object, but managed to achieve high mean class accuracy. Usually, convolutional neural networks are used for these 2D image classification techniques. In Ma et al. (2018), the authors have used a recurrent neural module along with CNNs. Hypergraph learning has been highly effective for object classification, as shown in Feng et al. (2019). One of the highest accuracy yielding approaches, RotationNet Kanezaki et al. (2019), also uses multiple views of the objects, albeit these are unsupervised viewpoints. Point-based approaches are most popular - they take raw point clouds as inputs and learns from their unstructured format, which makes them robust against corruption Qi et al. (2017a;b); Li et al. (2018b). One of the pioneering works in this direction is PointNet Qi et al. (2017a). PointNet++ Qi et al. (2017b) improved upon PointNet by capturing local geometric structures. Transformer-based learning strategies have received attention as well Zhao et al. (2021). Graph neural networks have been successful in classifying point cloud objects Wang et al. (2019); Mohammadi et al. (2021). Unlike these, our novel methodology extracts topological features, which are learned by a 1D CNN for object classification and achieves SOTA accuracy.

This paper uses TDA features extracted from the point clouds for object classification. Such a TDA-based approach has been previously used for MNIST data classification Garin & Tauzin (2019). TDA has recently been used to solve a diverse range of problems, such as in medical imaging Singh et al. (2023), biomedicine Skaf & Laubenbacher (2022), oncology Bukkuri et al. (2021), and cybersecurity Akcora et al. (2020).

## 2 DESCRIPTION OF TACO-NET

The TDA front-end of TACO-Net is a *universal, non-learning, dataset-agnostic pipeline* that can process any input point cloud by voxelizing it within its own axis-aligned bounding box (AABB), producing a binary cubical complex independent of global pose or scale normalization. The voxel grid is then transformed via a symmetric family of task-agnostic 3D filtrations (height, radial, density, erosion, dilation, and signed distance), each designed to probe complementary aspects of the cubical topology, yielding grayscale volumes on which cubical persistent homology is computed. From each persistence diagram, we extract stable topological summary statistics, including persistent entropy and multiple amplitudes (Wasserstein, Bottleneck, Betti curve, persistence landscape, and heat kernels), whose stability and injectivity properties ensure robust, shape-sensitive encodings. The pipeline, therefore, acts as a universal topological *shape sensor*, while the subsequent lightweight 1D CNN learns higher-order interactions over these fixed signatures to produce highly class-separable representations.

**Filtration types** Garin & Tauzin (2019). Let $\mathcal{B} : I \subseteq \mathbf{Z}^3 \to \{0, 1\}$ be a 3D binary image, where every $p \in I$ is a voxel. A voxel is *activated* if its value is 1; otherwise, it is *deactivated*. A grayscale filtration converts $\mathcal{B}$ into a grayscale 3D image $\mathcal{G} : I \subseteq \mathbf{Z}^3 \to \mathbf{R}$. Such filtrations can highlight different topological features in the binary image, even visually. We briefly describe the six kinds of filtrations used in TACO-Net to obtain a set of grayscale 3D images for every 3D binary image (constructed for every point cloud) for extracting topological feature vectors. Owing to the difference in the filtration functions, every filtration tends to highlight different features of $\mathcal{B}$. Refer to Fig. 4 in the Appendix for an illustration.

*Height filtration.* It needs a direction vector $v$ in 3-space. Every activated voxel $p \in \mathcal{B}$ is assigned a grayscale value that equals the distance between $p$ and the hyperplane defined by $v$. Every deactivated voxel is assigned the maximum distance between any voxel of $\mathcal{B}$ and the hyperplane defined by $v$, plus one. For TACO-Net, we have considered the 26 direction vectors in $\{\{0, 1, -1\}^3\} \setminus \{(0, 0, 0)\}$.

*Radial filtration.* A reference voxel $c$, called *center*, is supplied. Every activated voxel $p \in \mathcal{B}$ is assigned the distance between $p$ and $c$. The deactivated voxels are assigned the maximum distance between $c$ and any voxel, plus one. For `TACO-Net`, 27 centers $c_1, c_2, \ldots, c_{27}$ have been considered, chosen as the 27 vertices of a $3 \times 3 \times 3$ grid $\Xi_\mathcal{B}$ inside $\mathcal{B}$. We note that $\mathcal{C}_1 := [c_1, \ldots, c_9]$ belong to the first vertical slice of $\Xi_\mathcal{B}$ having the lowest $x$-coordinate, $\mathcal{C}_2 := [c_{10}, \ldots, c_{18}]$ the median, and $\mathcal{C}_3 := [c_{19}, \ldots, c_{27}]$ the highest. The centers are sorted lexicographically in every $\mathcal{C}_i$. Refer to Fig. 2 for an example.

The strength of our 26 height directions is that directional height filtrations discretize the Persistent Homology Transform Turner et al. (2014), which is injective on a broad class of shapes; hence, in principle, the family of height persistence diagrams determines the underlying shape with high accuracy, as shown later empirically. Our cube-symmetric set of 26 directions provides an efficient spherical sampling that harmonizes with cubical complexes, producing salient and stable topological events that differ across object categories. By the stability of persistent homology, these diagrams are robust to noise. Complementing height with a set of 27 radial filtrations, from carefully placed centers, and the following four filtrations, injects information about interior organization, yielding consistent gains. This is corroborated by the high accuracy numbers obtained for shape classification (see Sec. 3).

*Density filtration.* Every voxel $p \in \mathcal{B}$ is assigned a grayscale value equal to the number of activated voxels within a ball centered at $p$ having radius $r$. We fixed $r$ to 1 in our experiments.

*Dilation filtration.* Every voxel $p \in \mathcal{B}$ is assigned a grayscale value equal to the smallest Manhattan distance to an activated voxel in $\mathcal{B}$. Consequently, active voxels are assigned a 0 grayscale value.

*Erosion filtration.* It does the opposite of the dilation filtration. The dilation filtration is applied to the binary image $\mathcal{B}'$, obtained from $\mathcal{B}$ by changing activated voxels to deactivated and deactivated ones to activated. Deactivated voxels are assigned a 0 grayscale value.

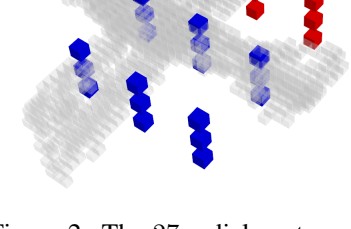

Figure 2: The 27 radial centers are shown for an airplane 3D binary image. For ModelNet40 and ModelNet10, we have used the centers $c_1, \ldots, c_{18}$, as shown in blue.

*Signed distance filtration.* For every activated voxel $p \in \mathcal{B}$, its grayscale value is the minimum Manhattan distance between $p$ and any deactivated voxel in $\mathcal{B}$ minus 1. For every deactivated voxel, its grayscale value is the negative of the minimum Manhattan distance between $p$ and any activated voxel in $\mathcal{B}$.

**Synergy between the filtrations.** Rather than treating filtrations as isolated tools, `TACO-Net` leverages their complementary strengths to build a unified pipeline. Height filtrations encode global shape cues by discretizing the Persistent Homology Transform, providing directional sensitivity and robustness to pose variations. Radial filtrations complement this by capturing interior organization and symmetry, injecting structural information that height alone cannot provide. Further shape understanding is achieved through four additional filtrations: density, dilation, erosion, and signed distance (collectively DEDS), each contributing unique geometric insights. Density filtration emphasizes local clustering by assigning grayscale values based on neighborhood occupancy, making it effective for distinguishing compact versus elongated shapes. Dilation filtration measures proximity to active voxels, highlighting external boundaries and object thickness, while erosion filtration does the opposite, revealing hollow regions and internal voids. Finally, signed distance filtration encodes both interior and exterior distances with positive and negative values, producing a nuanced representation of boundary smoothness and depth. This interplay enables TACO-Net to learn a holistic representation of 3D objects, making it robust to noise and generalizable across diverse datasets.

TDA Chazal & Michel (2021); Wasserman (2018) can extract topological information and geometric patterns from datasets using algebraic topology. Persistent homology Edelsbrunner et al. (2002); Zomorodian & Carlsson (2004) is a popular tool in TDA, applied to obtain different kinds of topological feature vectors of point clouds. It helps to understand the shape of a point cloud by tracking the birth and death of various

topological features (different from feature vectors), such as connected components, holes, and higher-dimensional voids that persist at different scales during an iterative process known as *filtration* (distinct from the six types of filtration mentioned above). A series of nested geometric structures is obtained at various scales during filtration. The topological features that persist (survive) across several iterations of a filtration, inside various sequences of such nested structures, can be used as topological descriptors to compute different topological feature vectors of a point cloud. For a comprehensive overview on persistent homology and various filtration techniques, we urge interested readers to refer to Chazal & Michel (2021); Edelsbrunner et al. (2002); Wasserman (2018); Zomorodian & Carlsson (2004). We use cubical homology and persistence meant for extracting topological information from cubical complexes.

**Cubical homology and persistence**. Kaczynski et al. (2006); Wagner et al. (2011) A finite *cubical complex* in 3-space is a union of points, line segments, squares, 3D cubes, aligned on the grid $\mathbf{Z}^3$. We leverage *cubical homology*, a variant of persistent homology meant for cubical complexes, to obtain topological feature vectors of grayscale 3D images, which are obtained from 3D binary images, constructed for every point cloud. Any grayscale 3D image can be perceived as a cubical complex $K$, where every voxel (a pixel in 3D) is a cube with an intensity value. The voxels, square faces of voxels, edges, and vertices are $3, 2, 1, 0$-cube, respectively. Hence, cubical homology can be applied directly to 3D grayscale images because of their natural grid-like structures. During filtration, the voxels are added in order of increasing intensity, forming a sequence of nested cubical subcomplexes. Starting with the lowest voxel intensity, all voxels with intensity at most $t$ are added to the current cubical complex along with their faces, edges, and vertices, where $t$ is the current voxel intensity being considered. A cubical complex obtained at step $i$ is a subcomplex of the complex obtained at step $i + 1$. Thus, we get a sequence of nested subcomplexes $K_0 \subseteq K_1 \subseteq \ldots \subseteq K_m$. Every $K_i$ is called a *sublevel set* of $K$, the cubical complex built from a given 3D image, since $K_i \subseteq K$. As voxels are added, the topological features, connected components of voxels (homology group $H_0$), tunnels or loops (homology group $H_1$), and enclosed cavities (homology group $H_2$), take birth or die. Every birth and death of a feature introduces a new birth-death pair in the *cubical persistence*, a multiset of points in $\mathbf{R} \times (\mathbf{R} \cup \{+\infty\})$, where every pair $(b, d)$ in the multiset denotes the birth of a topological feature at time $b$ and its death at time $d$. Long-surviving features are likely the significant features that can be used in classification tasks. Persistence, represented using a 2D scatter plot, is known as a *persistence diagram* (refer to Fig. 1). In a persistence diagram, every birth-death pair corresponds to a point in the diagram. The cubical persistence of a cubical complex is its *topological signature*. Next, we discuss the topological vectorization methods used here.

**Persistent entropy.** Chintakunta et al. (2015) Given a cubical persistence, $X = \{(b_i, d_i)\}$, its persistent entropy, denoted by $\rho(X)$, is a real number defined by as, $\rho(X) = -\sum_i p_i \log(p_i)$, where $p_i = \frac{d_i - b_i}{\ell(X)}$, and $\ell(X) = \sum_i (d_i - b_i)$. Having its roots in information theory, it gives an intuitive sense of disorder or complexity in the topological structure. We note that 3 real numbers are obtained for the 3 homology groups, $H_0, H_1,$ and $H_2$.

**Amplitude.** Introduced in Garin & Tauzin (2019), *amplitude* of a cubical persistence is defined as its distance to the empty persistence (devoid of birth-death pairs). It is used to compare two cubical persistences, obtained from two different 3D grayscale images. `TACO-Net` uses five types of amplitudes with varied parameters. Out of the five, two are metric-based (the Wasserstein and Bottleneck distances), and the remaining three are kernel-based (Betti curve, persistence landscape, and heat). For the Betti curve, and persistence landscape, the diagrams are sampled using 100 filtration values, whereas for the heat kernel, 20 are used. Let $X = \{(b_i, d_i)\}$ be a cubical persistence. For insight into the different kinds of amplitudes, we refer the reader to the Appendix.

*p-Wasserstein* Tauzin et al. (2021). The *half-lifetime* of a pair $(b_i, d_i) \in X$ is defined as $\frac{d_i - b_i}{2}$. The Wasserstein amplitude of order $p$, denoted by $W(X, p)$, is defined as the $L_p$ norm of the vector of half-lifetimes of the birth-death pairs in $X$. Hence, $W(X, p) = (\sum_i \left(\frac{d_i - b_i}{2}\right)^p)^{1/p}$. For `TACO-Net`, we have used $p = 1, 2$. We obtain 6 real numbers for this metric, since there are 3 homology groups and 2 values of $p$.

*Bottleneck* Tauzin et al. (2021). The Bottleneck amplitude is denoted by $B(X) = W(X, \infty)$. We obtain 3 real numbers for this metric due to the three homology groups.

*Betti curve. Tauzin et al. (2021)* The Betti curve of $X$ is the function $B_C : \mathbf{R} \to \mathbf{N}$, such that $B_C(s)$ gives the number of birth-pairs in $X$ that contains $s$ when every pair $(b_i, d_i)$ in $X$ is treated as an interval.

$$f_{(b_i, d_i)}(x) = \begin{cases} 0 & \text{if } x \notin (b_i, d_i) \\ x - b_i & \text{if } x \in (b_i, \frac{b_i + d_i}{2}] \\ -x + d_i & \text{if } x \in (\frac{b_i + d_i}{2}, d_i) \end{cases} \quad (1)$$

Two amplitudes are obtained using the $L_1$ and $L_2$ norms. We obtain 6 real numbers for this metric, since there are three homology groups and two norms.

*Landscape* Bubenik & Dłotko (2017); Bubenik et al. (2015). For a birth-death pair $(b_i, d_i) \in X$, let $f_{(b_i, d_i)} : \mathbf{R} \to [0, \infty]$, be a piecewise linear function given in Eq. 1. The *persistence landscape* of $X$ is the sequence of functions $\lambda_k : \mathbf{R} \to [0, \infty], k = 1, 2, 3, \ldots$ where $\lambda_k(x)$ is the $k$-th largest value of $\{f_{(b_i, d_i)}(x)\}_i$. Further, $\lambda_k(x)$ is set to 0 if the $k$-th largest value does not exist. The parameter $k$ is called the *layer*. For `TACO-Net`, we have used $k = 1, 2$. Four amplitudes are obtained using $L_1$ and $L_2$ norms for both the values of $k$. We get 12 real numbers for this metric, since there are three homology groups, two norms, and two distinct values of $k$.

*Heat kernel* Reininghaus et al. (2015). Gaussians of standard deviation $\sigma$ are placed over every point in $X$ and a negative Gaussian of $\sigma$ on the mirror point across the diagonal line in the persistence diagram. Thus, a real-valued function is obtained on $\mathbf{R}^2$. For `TACO-Net`, we have used $\sigma = 0.15$. We get 6 real numbers for this metric, since there are three homology groups and two norms, $L_1, L_2$.

Hence, for a given 3D grayscale image, obtained by using a filtration, we get a feature vector of length $3 + 33 = 36$, wherein 3 numbers are obtained using persistent entropy and the remaining 33 using amplitude.

**Feature selection and generation.** Refer to Fig. 5 in Appendix for an illustration. Let $P$ be a point cloud describing some 3D object. We convert $P$ into a voxelized 3D binary image $\mathcal{B}$ such that every active voxel contains at least one point from $P$. The volume of $\mathcal{B}$ is roughly equal to that of the axis-parallel bounding box of $P$. We run 57 filtrations on $\mathcal{B}$ yielding a set of 57 grayscale images. Out of 57 filtrations, there are 26 height filtrations for the 26 direction vectors in $\{\{0, 1, -1\}^3\} \setminus \{(0, 0, 0)\}$; 27 radial filtrations for the 27 centers $c_1, \ldots, c_{27}$, as described in Section 2; one each for the four types: density, dilation, erosion, and signed distance. As explained before, we extract 36 features from a grayscale image. Hence, due to the 57 filtrations used, which resulted in 57 binary images, the length of the final feature vector for $P$ is $57 \cdot 36 = 2052$. However, our experiments found that depending on the dataset, we must discard some of the radial filtrations from the initial 27 centers, as shown in Fig. 2 to achieve the highest possible accuracy.

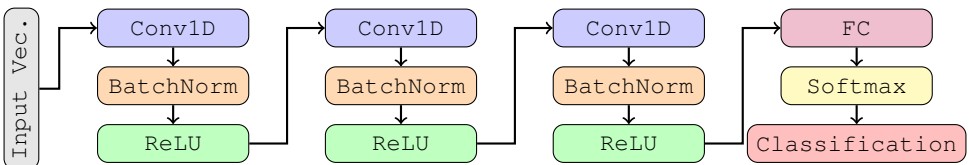

Figure 3: The architecture of the 1D CNN used in `TACO-Net`. Convolutional layers process the input feature vector before final classification via fully connected (FC) and softmax layers.

**Classification using a 1D CNN.** We use a lightweight 1D CNN deep neural network to classify the features extracted from the point clouds of the objects. In recent years, 1D CNN has been used extensively for such feature and sequence classification with high success Kiranyaz et al. (2021). Our network has three 1D CNN layers, each followed by batch normalization and ReLU layers. The three CNN layers after the input have filter sizes of $3, 5,$ and $7$ respectively, whereas the number of filters in the first two layers is

128 and 64, respectively. In the third CNN layer, the filters are set to the class count for ModelNet40 and ModelNet10 datasets, 40 for ScanObjectNN, and 32 for the VesselMNIST3D and AdrenalMNIST3D datasets.

After the three consecutive 1D CNN layers, we have a fully connected layer of size equal to the number of classes. Next, the classification is done by applying a softmax function on the outputs of the fully connected layer. Refer to Fig. 3. The time taken for classification is $\mathcal{O}(n + v^3 + v/\rho^3)$, where $n$ is the size of the point cloud, $v$ is the number of voxels in $\mathcal{B}$, and $\rho$ is the voxel-size used. See Appendix for a proof.

Table 1: Parameters and values

| Parameters | Values |
| --- | --- |
| Max. training epoch | 1000 |
| Loss stop threshold | 0.005 |
| Learning rate | 0.001 |
| Minibatch size | 128 |
| Optimizer | Adam |
| Voxel size | 0.05 |

## 3 EXPERIMENTS

**Settings.** We have used 7 datasets to validate the efficacy of `TACO-Net` and they are ModelNet10/40 (by far the most popular benchmark for this problem), OmniObject3D (a real-world dataset consisting of 190 classes), ScanObjectNN (a real-world noisy dataset with 15 classes), 3DGrocery100 (a large dataset of grocery items), and two real-world binary medical datasets for further generalizability, namely VesselMNIST3D and AdrenalMNIST3D. More details about these datasets are provided in the Appendix (Sec. 4.3.1).

We have implemented all TDA-related portions of `TACO-Net` in Python using the `giotto-tda` package Tauzin et al. (2021). The experiments were run on a machine equipped with an Intel `i9-12900K` processor, 32-GB of main memory, and a NVIDIA RTX 3060 GPU. For all the datasets, we evaluate the performance on two main metrics: overall accuracy (OA - average accuracy % across all test cases) and mean class accuracy (mAcc - mean accuracy % across all classes). These are the most common evaluation metrics in object classification. For VesselMNIST3D, we also present the F1-score metric due to imbalance.

For ModelNet40, we have found the vector length 1728 to be the optimum in terms of OA. The exact number of filtrations that corresponds to this length is 48, out of which 26 are height, one each from the four types: density, dilation, erosion, and signed distance, and 18 are radial corresponding to the 18 centers $c_1, \ldots, c_{18}$, as described in Section 2 and shown in Fig. 2. Therefore, all the results presented below are with 1728-length feature vectors. We have used the same vector length input for ModelNet10 as well. The optimum length for VesselMNIST is 1152 using the two centers $c_1, c_2$, and for AdrenalMNIST it is 1584 using $c_1, \ldots, c_{14}$. Other relevant parameters and their values for TDA-related experiments are mentioned in Section 2.

We used MATLAB to implement the 1D CNN. We stopped our training early if the training loss had reached 0.005. Each configuration has been trained 5 times, and the average results are presented in the paper unless specified otherwise. The learnable parameters for ModelNet40, ModelNet10, VesselMNIST, and AdrenalMNIST are 0.72M, 0.71M, 0.50M, 0.66M, respectively. The parameters used in our experiments and their values are listed in Table 1.

### 3.1 RESULTS

**ModelNet10/40.** First, we present the results of testing `TACO-Net` on ModelNet40 and 10 datasets. To begin with, we first illustrate the empirical reason behind choosing 18 radial filtration centers along with DEDS and height filters for the ModelNet40 dataset. This result is presented in Fig. 8(a). As can be seen, with feature vector length 1728, i.e., 18 radial filtration filters, the OA is the highest. Although with the different other feature lengths, the OA is close, but lower than the one with length = 1728. Next, we have tested `TACO-Net` with different voxel sizes to create the 3D binary image from the given point cloud. We have noticed that with a voxel size 0.05, the OA is the highest. With 0.03 and 0.07, the accuracy values decrease to 98.61 (OA), 96.96 (mAcc), and 98.30 (OA) and 96.16 (mAcc), respectively.

Next, the benchmark results for both 40- and 10-class variants of the ModelNet dataset are presented in Tables 2 and 4 (in the Appendix), respectively. These results prove that our proposed `TACO-Net` framework achieves

state-of-the-art OA and mAcc accuracies for both these datasets. Notably, `TACO-Net` achieves $1.68\%$ higher OA than RotationNet, the current highest OA-achieving method on the ModelNet40 dataset. Further, `TACO-Net` comprehensively outperformed the recent transformer-based models, e.g., PointMamba Liang et al. (2024) and PointGPT Chen et al. (2023), among others. Our macro-averaged precision-recall curve (Fig. 8(b)) stays tightly clustered near $(1, 1)$, showcasing near-perfect precision and recall across every class. This level of consistency, even on minority classes, sets a new bar for robust, balanced multi-class performance. Fig. 9 (refer to Appendix) shows the confusion matrix found with the best saved model.

Similarly, `TACO-Net` outperforms Rotation-Net in OA on the ModelNet10 dataset. Given the lower number of classes available, it was expected that the proposed `TACO-Net` framework would achieve higher accuracies in ModelNet10 than in ModelNet40. Not only was `TACO-Net` successful in meeting that expectation, it yielded $99.52\%$ OA and mAcc values. Most importantly, to the best of our knowledge, ours is the first approach to push the classification accuracy beyond $99\%$ on both Model-Net40 and ModelNet10. Altogether, these results make this research work groundbreaking.

**Real-world Datasets.** To further validate real-world applicability, we evaluate `TACO-Net` on OmniObject3D Wu et al. (2023), a challenging benchmark featuring thousands of everyday objects captured under realistic conditions. Despite its complexity and large class diversity, `TACO-Net` delivers the highest accuracy of $58.90\%$, decisively outperforming heavyweight baselines including CurveNet, PointNet, PointNet++, and PCT: Point cloud transformer (see Table 5).

Table 2: Accuracy (%) results on ModelNet40.

| Method | OA | mAcc |
|---|---|---|
| 3DShapeNets Wu et al. (2015) | 84.7 | 77.3 |
| PointNet Qi et al. (2017a) | 89.2 | 86.2 |
| MVCNN Su et al. (2015) | 90.1 | - |
| Ma et al. Ma et al. (2018) | 91.05 | - |
| KD-Network Klokov & Lempitsky (2017) | 91.8 | 88.5 |
| PointNet++ Qi et al. (2017b) | 91.9 | - |
| PointCNN Li et al. (2018b) | 92.5 | 88.1 |
| OctFormer Wang (2023) | 92.7 | - |
| GVCNN Feng et al. (2018) | 93.1 | - |
| PointNeXt Qian et al. (2022) | 93.2 | 90.8 |
| PCM Zhang et al. (2025) | 93.4 | 90.7 |
| PointMamba Liang et al. (2024) | 93.6 | - |
| Point-Transformer Zhao et al. (2021) | 93.7 | 90.6 |
| Point-Bert Yu et al. (2022) | 93.8 | - |
| PointMLP Ma et al. (2022) | 94.5 | 91.4 |
| MHBN Yu et al. (2018) | 94.7 | 93.1 |
| PointGPT Chen et al. (2023) | 94.9 | - |
| Mamba3D Han et al. (2024) | 95.1 | - |
| Pointview-GCN Mohammadi et al. (2021) | 95.4 | - |
| VRN Ensemble Brock et al. (2016) | 95.54 | - |
| HGNN Feng et al. (2019) | 96.6 | - |
| RotationNet Kanezaki et al. (2019) | 97.37 | 96.29 |
| `TACO-Net` (Ours) | **99.05** | **97.97** |

We next evaluate `TACO-Net` on ScanObjectNN (all six popular variants), a notoriously challenging real-world benchmark characterized by heavy occlusions and background clutter-conditions where many point-based methods struggle Uy et al. (2019). We compare our numbers only against from-scratch supervised learning methods. We acknowledge that there are 'training from pre-training' methods that have achieved slightly better numbers than ours (e.g., PointMamba Liang et al. (2024)). Still, to maintain fairness, we have only compared against similar methods that do not use pre-trained models. First, we compare the OBJ_BG, OBJ_ONLY, and the hardest variant, PB_T50_RS, against existing techniques. These results are presented in Table 6. For the OBJ_BG and OBJ_ONLY variants, we achieved SOTA accuracy and second highest for PB_T50_RS, among from-scratch supervised learning methods as shown in Table 6 while outperforming seminal works such as PointNet Qi et al. (2017a), PointNet++ Qi et al. (2017b), DGCNN Wang et al. (2019), and PointNeXt Qian et al. (2022), among others, also outperforming some of the most recent studies such as ADS Hong et al. (2023) and X-3D Sun et al. (2024). Further, when tested on the other three variants (PB_T25, PB_T25_R, PB_T50_R) of ScanObjectNN, `TACO-Net` yields $\geq 91\%$ accuracy (see Table 7). When compared against XGBoost, `TACO-Net` achieves $\sim 30\%$ higher accuracy, which proves that the topological features are powerful, but the proposed 1D CNN truly learns descriptive patterns from them, unlike other approaches such as XGBoost. Overall, the ScanObjectNN results underscore `TACO-Net`'s ability to remain yield SOTA accuracy in highly cluttered, real-world scenarios with backgrounds and transformations.

`TACO-Net` achieved a competitive OA of 40.18% on the recent and challenging 3DGrocery100 dataset She-shappanavar et al. (2024), outperforming strong state-of-the-art methods, PointNet Qi et al. (2017a) and Point cloud transformer Guo et al. (2021), which achieved OAs of 37.61% and 39.67%, respectively.

**Real-world Medical Data.** For VesselMNIST Yang et al. (2020), we compared against some of the current SOTA benchmarks for this dataset as shown in Table 8. When compared against the benchmarks presented in Yang et al. (2020), `TACO-Net` performed better in terms of both F1 and mAcc. For example, the previous highest F1-score of 0.90 for VesselMNIST was achieved by PointNet++ and PointCNN, whereas our mean F1-score is 0.94 - an improvement of 4.44%. Similarly, for the mAcc metric, our average result is 95.28%, whereas the prior best was 93.52 achieved by PointNet++ - an improvement of 1.88%.

For the AdrenalMNIST dataset, we used the benchmark provided in Yang et al. (2023) as our baseline. The comparison results are presented in Table 9. The authors in Yang et al. (2023) have used different variations of ResNet along with medical image-specific variants such as ACS Yang et al. (2021). Our proposed `TACO-Net` outperformed all these benchmarks in the OA metric, as shown in the table. Notably, for the VesselMNIST3D dataset, the difference in OA between ours and the current SOTA achieved using ResNet-18 + ACS is 4.58%.

**Resiliency Against Corrupted Test Data.** Noise resiliency in `TACO-Net` is achieved because topological features, extracted via persistent homology, capture the global shape characteristics of objects rather than relying on exact point positions. These features remain stable under small perturbations or noise, as persistent homology emphasizes long-lived topological structures while ignoring short-lived, noise-induced artifacts. Consequently, `TACO-Net` maintains high accuracy in most cases even when point clouds are corrupted, as discussed below. To test the robustness of `TACO-Net`, we have used the standardized approach of ModelNet40-C Sun et al. (2022), where the test set of ModelNet40 is perturbed by adding various common types of corruptions. Note that the 'uniform downsampling' perturbation was not part of ModelNet40-C.

Two main differences between our implementation and ModelNet40-C are 1) we use $2048 \times 3$ size point clouds, and 2) we do not apply any normalization after the perturbation is incorporated. The results for this study are presented in Table 10. The first row presents the test results found by the model with the highest test accuracy for clean ModelNet40 without any corruption. We use this saved model for inference on the corrupted test dataset and report the results in Table 10. Two severity levels of data corruption are chosen from Sun et al. (2022): 1 and 5, named Low and High, respectively, in our paper. For 'uniform downsampling', we have removed 10 and 30% random points uniformly for low and high severity, respectively.

We see that `TACO-Net` is very robust against low-level perturbations - always achieves $\geq 94\%$ OA except for 'impulse', where the OA falls to 52.88%. As expected with high-level perturbations, `TACO-Net` shows resiliency. In case of 'impulse', the OA more than halves, but for the others, it performs reasonably well - the average OA being 68.65%. If the underlying shape changes significantly, then topological features become very different from clean class signatures, and `TACO-Net` struggles to recognize corrupted point clouds. Under the high-severity corruption, augmenting the training set with corrupted copies of randomly selected 20% of training instances lifts overall accuracy to 96.11% in the case of rotation, for example, substantially outperforming the non-augmented model (49.39%). This demonstrates that a task-aligned augmentation confers significant robustness to extreme pose variations without altering the core architecture.

**Shape Retrieval.** Our method achieves a new state-of-the-art retrieval performance with an mAP of 99.33 on ModelNet40, surpassing the strongest baseline Latformer (97.4) and all prior approaches (see summary results in Table 13 in the Appendix). Beyond accuracy, the framework demonstrates strong generalizability, showing its effectiveness not only for classification but also for challenging tasks such as shape retrieval.

**Note on Efficiency.** With voxel size $\rho = 0.05$ (our optimal configuration), the throughput of the TDA feature generation pipeline is 5.3 point clouds/second, when the feature vector length was set to 1728 for the ModelNet40 dataset. This number increased to 8.2 and decreased to 1.4 when $\rho$ was increased to 0.07 or decreased to 0.03, respectively. However, as mentioned earlier, the test accuracy decreases in both cases.

However, on the bright side, due to the lightweight 1D CNN of `TACO-Net`, the training time is short (2.50 mins.) and class prediction is lightning-fast – achieving a throughput rate of $16,454$ point clouds/second. Furthermore, Table 12 demonstrates that our proposed `TACO-Net` model achieves state-of-the-art accuracy with the very few parameters (0.72M), highlighting its superior efficiency compared to prior methods.

**Ablation Study.** Our approach to studying the effect of ablation is two-fold. First, using our proposed network (Fig. 3), we test different sets of topological features extracted from a point cloud, i.e., using density, erosion, dilation, and signed distance filtration (DEDS) only, height (H) filtration only, and finally DEDS + H only (i.e., without any radial filtration). The effect of using different radial filtration along with DEDS and H together is already illustrated in Fig. 8(a) and discussed earlier. In the next set of ablation studies, we use the best topological features for ModelNet40, i.e., a vector length of $1728$, while testing the effect of deleting one CNN layer at a time. The results are listed in Table 3. This study demonstrates that the features extracted via the H filtration are the most effective, yielding over 98% OA, which outperforms all baselines while being 3.5x faster than the whole pipeline. Similarly, when just entropy is used (without amplitude), vector generation is 2.2x faster while maintaining $96.16\%$ OA. This shows `TACO-Net` is not only accurate but also tunable for resource-constrained scenarios. The finding further supports our rationale for employing 26 directional vectors, as outlined in Sec. 2. Incorporating the DEDS features provides a slight improvement, but the gain is marginal. In contrast, removing two CNN layers leads to a moderate decrease in 4.29% in accuracy. To highlight the effectiveness of our 1D CNN for feature vector classification, we replace it with a heavier 2.2M-parameter transformer that incorporates feature and positional embeddings, mixed self-attention, and a fully connected classifier. Despite its complexity, this transformer yields only $62.20\%$ accuracy on ModelNet40, underscoring the superiority of our lightweight CNN design. On the other hand, XGBoost, a non-deep learning method, also yielded substantially lower OA of 81.35% (see Table 11). Taken together, these results provide strong justification for our feature selection and network design choices.

**t-SNE Analysis on ModelNet40.** Remarkably, using only non-learned topological descriptors as input, the proposed 1D CNN model produces t-SNE embeddings with clear class-wise separation, indicating that the network effectively distills these signatures into robust, discriminative feature representations (Fig. 10). This shows the strong compatibility of topological features with modern deep architectures.

## 4 CONCLUSION AND FUTURE WORK

Computer vision researchers have made significant advancements in 3D object classification using various deep learning techniques in recent years. However, there are still some challenges to address and open directions to explore. To this end, we have proposed a novel framework, named `TACO-Net`, for 3D object classification. Our proposed approach takes a point cloud of the object as input, converts it into a voxelized 3D binary image, extracts topological signatures from it through various filtration techniques, and finally learns these features using a lightweight 1D CNN. Results show that our proposed technique achieves near-perfect overall accuracy in popular 3D object classification benchmark datasets, namely ModelNet40 and ModelNet10, while outperforming the current SOTA for these. Further, when tested on two 3D medical datasets consisting of brain MRA and abdominal CT scan data, `TACO-Net`, outperforms all the benchmarks provided in the literature, showcasing its strong generalizable qualities. To enhance real-time performance, future work will focus on exploiting GPU parallelism to increase throughput for the topological feature generation pipeline.

Table 3: Ablation study on ModelNet40.

| | Variant | OA | mAcc |
|---|---|---|---|
| features | DEDS only | 96.52 | 93.76 |
| | Density only | 80.75 | 75.59 |
| | E only | 52.26 | 43.14 |
| | Dilation only | 85.68 | 81.61 |
| | S only | 89.35 | 81.10 |
| | R only | 95.47 | 91.52 |
| | H only | 98.29 | 96.38 |
| | DEDS + H only | 98.82 | 97.33 |
| | Entropy only | 96.16 | 92.79 |
| net | First 2 Conv1D | 98.18 | 95.93 |
| | First Conv1D | 94.76 | 90.70 |

## ETHICS STATEMENT

This research poses minimal ethical risks, as it uses only publicly available benchmarks and anonymized medical datasets. The work does not involve the collection of sensitive or personally identifiable data. Its potential positive impacts include improving autonomous navigation, robotics, and medical diagnostics. However, as with any object recognition system, there is a possibility of misuse in surveillance or military applications, which warrants awareness and responsible deployment. Overall, the benefits to safety and healthcare outweigh foreseeable risks.

## REPRODUCIBILITY STATEMENT

We have provided the Python and MATLAB codebase as supplementary material in a ZIP file. The best model (in MATLAB) has also been included for reproducing our numbers. The dataset details are provided in the Appendix.

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

## APPENDIX

### 4.1 TACO-NET PIPELINE AND ILLUSTRATION OF CUBICAL PERSISTENCE

We present an illustration of various filtrations used in TACO-Net in Fig. 4.

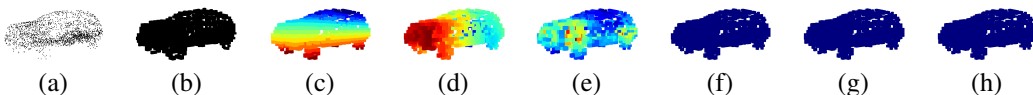

(a)     (b)     (c)     (d)     (e)     (f)     (g)     (h)

Figure 4: A 2048-element point cloud of a car from ModelNet40 is shown in (a) and its 3D binary image with voxel size 0.05 in (b). The grayscale images obtained after height using $v : (-1, 0, 0)$, radial using $c : (4, 4, 10)$, density, dilation, erosion, and signed distance filtrations, are shown in (c), (d), (e), (f), (g), (h), respectively. Hotter voxels have higher grayscale values. For brevity, voxels outside the shape are not shown; consequently, (f), (g), and (h) appear almost the same.

We present a diagram of the TACO-Net pipeline in Fig. 5.

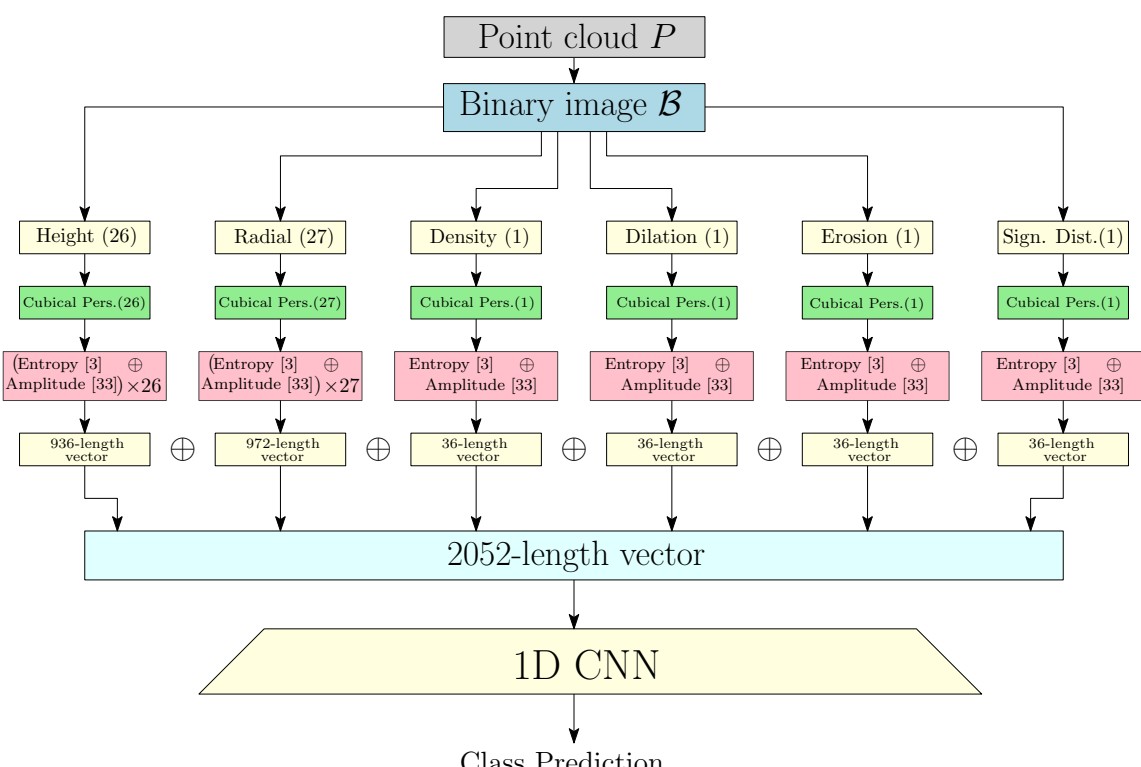

Figure 5: An illustration of the novel pipeline of TACO-Net. The numbers in the parentheses denote the number of variants. For instance, 'Height (26)' implies that 26 variants of the height filtration have been used. The integers inside the square braces denote vector length. For example, Entropy[3] implies that applying entropy to cubical persistence yields a vector of length 3. Further, $\oplus$ denotes vector concatenation.

**An illustrative example.** We give an example of cubical persistence in Fig. 6(a-d).

A 2D grayscale image with pixels having their grayscale values in $\{0, 50, 100\}$ is shown in (a). During filtration, 0-pixels are considered first, then 50-pixels, and finally 100-pixels. In (b), $K_0$ is shown; the 0-pixels are added, resulting in three connected components, each comprising just one pixel. (c) $K_1$: The 50-pixels are added. Consequently, there is just one connected component that looks like the digit 6. In the previous step, there were three, but now just one. So, three connected components took birth at 0, and two of them died at 50. A hole takes birth inside $K_1$. (d) $K_2$: The 100-pixels are added. The hole obtained in the previous step dies in this step.

In the homology dimension 0, there are two birth-death pairs $(0, 50), (0, 50)$ since two connected components died, and in dimension 1, there is exactly one $(50, 100)$ since the hole formed at 50 and died at 100. Hence, the persistence (a multiset) contains three pairs. This is expressed pictorially in Fig. 7. In the end, there is just one connected component that never dies.

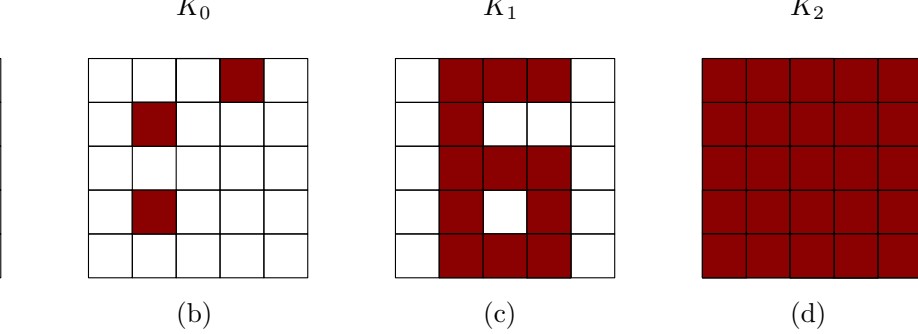

Figure 6: Illustrating filtration for cubical persistence (here shown in 2D). Note that $K_0 \subseteq K_1 \subseteq K_2$.

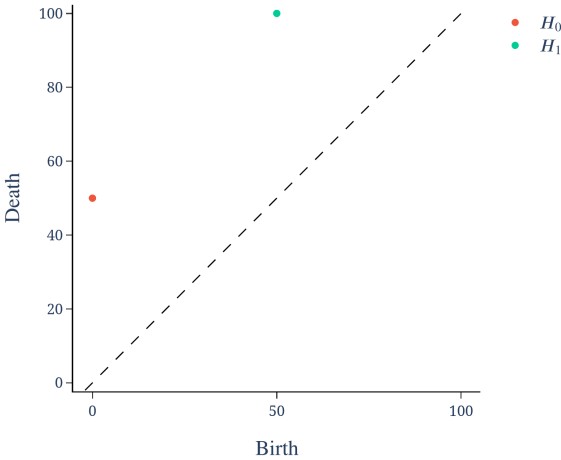

Figure 7: The persistence diagram corresponding to the filtration shown above. There are two overlapping red dots for the two birth-death pairs $(0, 50), (0, 50)$. The green dot corresponds to the pair $(50, 100)$.

**Rationale for the Amplitude Kinds.**

In what follows, we provide an expansion on the amplitude discussion presented in Sec. 2, to provide insights on their use.

*p-Wasserstein* Tauzin et al. (2021). The *half-lifetime* of a pair $(b_i, d_i) \in X$ is defined as $\frac{d_i - b_i}{2}$. The Wasserstein amplitude of order $p$, denoted by $W(X, p)$, is defined as the $L_p$ norm of the vector of half-lifetimes of the birth-death pairs in $X$. Hence, $W(X, p) = (\sum_i (\frac{d_i - b_i}{2})^p)^{1/p}$. For TACO-Net, we have used $p = 1, 2$. We obtain 6 real numbers for this metric, since there are 3 homology groups and 2 values of $p$. This metric aids in measuring the $L_p$ norm of half-lifetimes, providing a stable (to diagram perturbations) and tunable sensitivity to feature persistence. Using $p = 1$ emphasizes the aggregate contribution of many moderate features, while $p = 2$ weights longer lifetimes, more strongly complementary views that improve discrimination.

*Bottleneck* Tauzin et al. (2021). The Bottleneck amplitude is denoted by $B(X) = W(X, \infty)$. We obtain 3 real numbers for this metric due to the three homology groups. This metric captures the single most persistent topological structure, which often aligns with the dominant, category-defining shape cue. Its robustness to small noise and invariance to minor diagram perturbations make it a strong separator when a few long-lived features matter most.

*Betti curve. Tauzin et al. (2021)* The Betti curve of $X$ is the function $B_C : \mathbf{R} \to \mathbf{N}$, such that $B_C(s)$ gives the number of birth-pairs in $X$ that contains $s$ when every pair $(b_i, d_i)$ in $X$ is treated as an interval. Two amplitudes are obtained using the $L_1$ and $L_2$ norms. We obtain 6 real numbers for this metric, since there are three homology groups and two norms. This metric helps summarize "how many features are alive" across filtration values, yielding an interpretable 1D profile of topology over scale. The norms over this curve provide compact, stable vectors that are efficient to learn with a 1D CNN while retaining multi-scale counts.

*Landscape* Bubenik & Dłotko (2017); Bubenik et al. (2015). For a birth-death pair $(b_i, d_i) \in X$, let $f_{(b_i, d_i)} : \mathbf{R} \to [0, \infty]$, be a piecewise linear function given in the following equation.

$$f_{(b_i, d_i)}(x) = \begin{cases} 0 & \text{if } x \notin (b_i, d_i) \\ x - b_i & \text{if } x \in (b_i, \frac{b_i + d_i}{2}] \\ -x + d_i & \text{if } x \in (\frac{b_i + d_i}{2}, d_i) \end{cases}$$

The *persistence landscape* of $X$ is the sequence of functions $\lambda_k : \mathbf{R} \to [0, \infty], k = 1, 2, 3, \dots$ where $\lambda_k(x)$ is the $k$-th largest value of $\{f_{(b_i, d_i)}(x)\}_i$. Further, $\lambda_k(x)$ is set to 0 if the $k$-th largest value does not exist. The parameter $k$ is called the *layer*. For TACO-Net, we have used $k = 1, 2$. Four amplitudes are obtained using $L_1$ and $L_2$ norms for both the values of $k$. We get 12 real numbers for this metric, since there are three homology groups, two norms, and two distinct values of $k$. This metric encodes order-statistics of feature prominence via layers $\lambda_k$, preserving more geometric detail than simple counts yet remaining Hilbert-space friendly for averaging and norms. Using $k = 1, 2$ captures dominant and secondary structures, offering a stable, rich functional summary that boosts class separability.

*Heat kernel* Reininghaus et al. (2015). Gaussians of standard deviation $\sigma$ are placed over every point in $X$ and a negative Gaussian of $\sigma$ on the mirror point across the diagonal line in the persistence diagram. Thus, a real-valued function is obtained on $\mathbf{R}^2$. For TACO-Net, we have used $\sigma = 0.15$. We get 6 real numbers for this metric, since there are three homology groups and two norms, $L_1, L_2$. This metric places (positive) Gaussians on diagram points and (negative) mirrors across the diagonal, yielding a smooth, multi-scale similarity that is robust to small birth/death shifts. This continuous embedding captures spatial arrangement in the diagram and works well with standard norms; our $\sigma = 0.15$ balances noise-tolerance and sensitivity.

### 4.2 THEORETICAL ANALYSIS

**Theorem 1.** *Let $P$ be an $n$-element point cloud that needs to be classified by* `TACO-Net`. *Then, the time taken for classification is $\mathcal{O}(n + v^3 + v/\rho^3)$, where $v$ is the number of voxels in $\mathcal{B}$ and $\rho$ is the voxel-size used.*

*Proof.* Initializing all voxels in $\mathcal{B}$ as inactive requires $\mathcal{O}(v)$ time. For each point in $P$, locating the corresponding voxel in $\mathcal{B}$ takes $\mathcal{O}(1)$ time. Since $|P| = n$, the total time to prepare $\mathcal{B}$ is $\mathcal{O}(v + n)$.

From $\mathcal{B}$, we generate 57 grayscale images using six filtration types: height, radial, density, dilation, erosion, and signed distance. For the 26 height and 27 radial filtrations, each voxel requires a constant-time distance computation, resulting in $\mathcal{O}(v)$ time per filtration. Thus, the total time for generating these 53 grayscale images is $\mathcal{O}(57 \cdot v) = \mathcal{O}(v)$, including initialization. For the density filtration, each voxel must inspect its neighborhood within a ball of radius $r$, which contains $\mathcal{O}(r^3/\rho^3)$ voxels. In `TACO-Net`, we set $r = 1$, yielding a per-voxel cost of $\mathcal{O}(1/\rho^3)$, and a total cost of $\mathcal{O}(v/\rho^3)$. The remaining three filtrations, dilation, erosion, and signed distance, can be computed in $\mathcal{O}(v)$ time each using efficient distance transform algorithms Fabbri et al. (2008). Therefore, the total time for generating all 57 grayscale images is $\mathcal{O}(v(1 + 1/\rho^3))$.

Cubical persistence for a single grayscale image can be computed in $\mathcal{O}(v^3)$ time using standard matrix reduction techniques Wagner et al. (2011). For 57 images, the total cost is $\mathcal{O}(57 \cdot v^3) = \mathcal{O}(v^3)$. Each voxel generates up to 27 cells (1 cube, 6 faces, 12 edges, 8 vertices), the building blocks of a cubical complex. Each cell can belong to at most one persistence pair. Hence, the worst-case number of birth–death pairs is at most $27v = \mathcal{O}(v)$.

Feature extraction from each cubical persistence involves computing persistent entropy and amplitude metrics. Persistent entropy requires $\mathcal{O}(v)$ time per image. Wasserstein and Bottleneck amplitudes also take $\mathcal{O}(v)$ time. The Betti curve kernel, evaluated over 100 filtration values, requires $\mathcal{O}(v)$ time per image. The persistence landscape kernel, which involves sorting at each of 100 sampled values, incurs $\mathcal{O}(v \log v)$ time. The heat kernel, evaluated over 20 filtration values, takes $\mathcal{O}(v)$ time. Thus, the total time to generate the topological features for one grayscale image is $\mathcal{O}(v \log v)$. For the 57 images, time taken is $\mathcal{O}(57 \cdot v \log v) = \mathcal{O}(v \log v)$.

Since the 1D CNN model is fixed during inference, classification of the topological vector takes constant time, i.e., $\mathcal{O}(1)$.

Combining all components, the overall time complexity of the pipeline is:

$$\mathcal{O}(v + n) + \mathcal{O}\left(v\left(1 + \frac{1}{\rho^3}\right)\right) + \mathcal{O}(v^3) + \mathcal{O}(v \log v) + \mathcal{O}(1) = \mathcal{O}\left(n + v^3 + \frac{v}{\rho^3}\right).$$

$\square$

### 4.3 FURTHER EXPERIMENTAL DETAILS AND RESULTS

#### 4.3.1 DATASETS

We have used the following six datasets to test the performance of the proposed `TACO-Net` framework.

- ModelNet40 Wu et al. (2015): It is one of the most popular benchmark datasets. The dataset comprises 40 classes, each consisting of CAD models of everyday objects. We used the official split, which consisted of 9,843 shapes for training and 2,468 for testing. For every shape, a random uniform sample of 2048 3D points was extracted from these CAD objects for the classification task.

- ModelNet10: A smaller, 10-class version of ModelNet40 Wu et al. (2015) with 3991 train and 908 test samples. Similar to ModelNet40, a 2048-element point cloud for each object was used in our experiments.

- OmniObject3D: A real-world point cloud object dataset, which is notoriously difficult to classify Wu et al. (2023). Every point cloud in the dataset has a size of 4096. With a large number of categories, it poses an extreme class-imbalance and inter-class similarity challenge, making accurate classification significantly harder compared to smaller-scale benchmarks. Unlike ModelNet10/40, there is no official train/test split available for this dataset. Therefore, we used an $80/20$ split, without instance leakage.

- ScanObjectNN: The ScanObjectNN dataset Uy et al. (2019) is a large-scale benchmark designed to evaluate 3D point cloud classification methods under realistic, noisy conditions. Unlike earlier synthetic datasets such as ModelNet40, ScanObjectNN is built from real-world scans captured by RGB-D sensors, making it far more representative of practical 3D perception challenges. It contains over 15,000 object instances across 15 categories, each featuring clutter, occlusion, background artifacts, and varying viewpoints. The dataset includes multiple variants. In our experiments, we have used the popular 6 variants - OBJ_ONLY, OBJ_BG, PB_T25, PB_T25_R, PB_T50_R, and PB_T50_RS. We have used the original 2048-element point clouds in our experiment.

- VesselMNIST3D: In Yang et al. (2020), the authors have introduced an open-access 3D intracranial aneurysm dataset with $103$ 3D meshes from brain Magnetic Resonance Angiography (MRA). This dataset has two classes: $1,694$ healthy vessel segments (V.) and $215$ aneurysm segments (A.). The dataset has the training, validation, and test set ratio of $7:1:2$ Yang et al. (2023).

- AdrenalMNIST3D: It is a CT scan dataset with two classes, consisting of shape masks from $1,584$ left and right adrenal glands (i.e., $792$ patients) Yang et al. (2023). The binarized images for the two medical datasets are provided through the `medmnist` Python package and have a resolution of $28 \times 28 \times 28$. Therefore, our starting point is 3D binary images instead of point clouds for AdrenalMNIST3D and VesselMNIST3D.

- 3DGrocery100: It is a comprehensive large-scale dataset designed for 3D object recognition in grocery environments Sheshappanavar et al. (2024) . It contains 87,898 3D point clouds extracted from over 10,000 RGB-D images and covers 100 categories of everyday grocery items, ranging from packaged foods to produce and household essentials. The dataset contains 66,032 training and 21,866 test 1024-element point clouds.

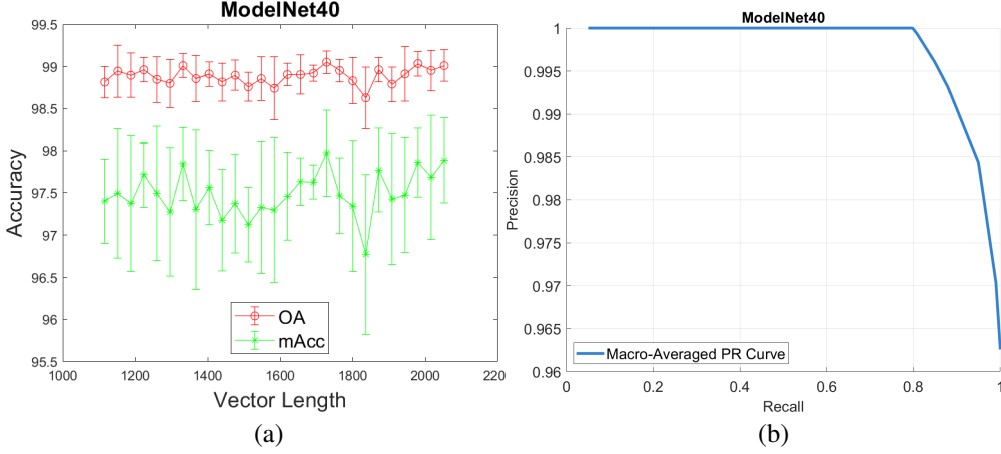

(a)                     (b)

Figure 8: ModelNet40: (a) Change in OA/mAcc w.r.t. feature vector length (i.e., different number of radial filtrations) and (b) Precision-recall curve for the best OA model.

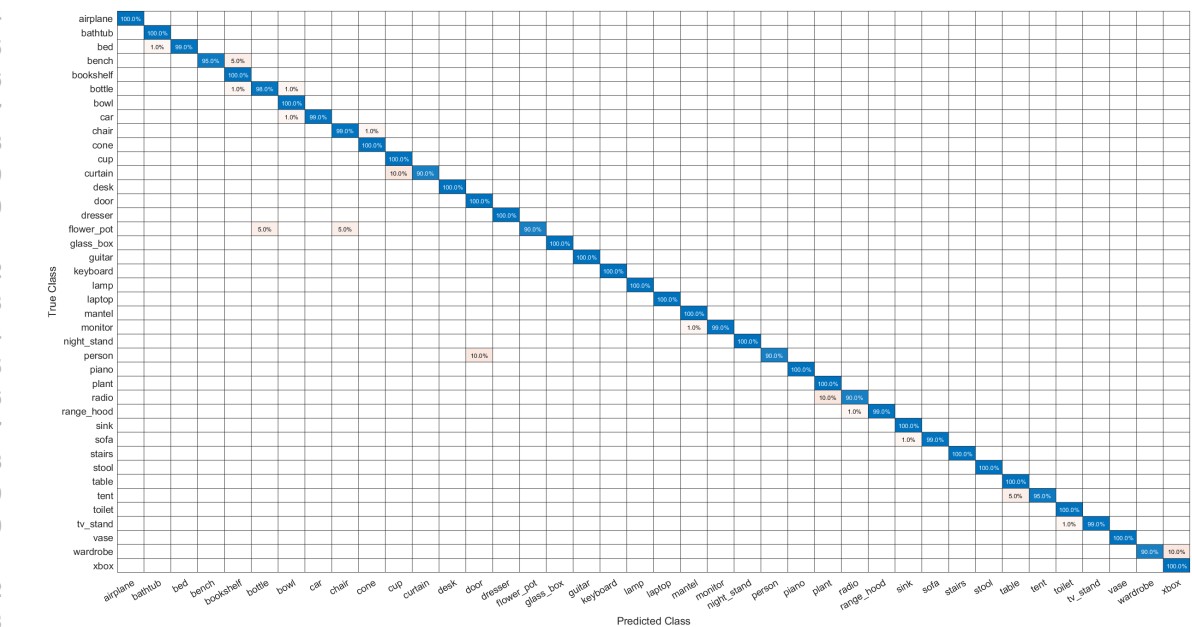

Figure 9: Confusion matrix for the best OA model.

Table 4: Classification accuracy (%) results on ModelNet10 dataset.

| ModelNet10 | | |
|---|---|---|
| Method | OA | mAcc |
| 3DShapeNets Wu et al. (2015) | 83.54 | - |
| 3D-GAN Wu et al. (2016) | 91 | - |
| VoxNet Maturana & Scherer (2015) | 92 | - |
| Primitive-GAN Khan et al. (2019) | 92.2 | - |
| ORION Sedaghat et al. (2017) | 93.9 | - |
| KD-Network Klokov & Lempitsky (2017) | 94 | 93.5 |
| MHBN Yu et al. (2018) | 95 | 95 |
| 3DmFV-Net Ben-Shabat et al. (2018) | 95.2 | - |
| Point2Sequence Liu et al. (2019a) | 95.3 | 95.1 |
| A-CNN Komarichev et al. (2019) | 95.5 | 95.3 |
| RCNet-E Wu et al. (2019) | 95.6 | - |
| PANORAMA-ENN Sfikas et al. (2018) | 96.85 | - |
| VRN Ensemble Brock et al. (2016) | 97.14 | - |
| Grid-GCN Xu et al. (2020) | 97.5 | 97.4 |
| RotationNet Kanezaki et al. (2019) | 98.9 | - |
| TACO-Net (Ours) | **99.52** | **99.52** |

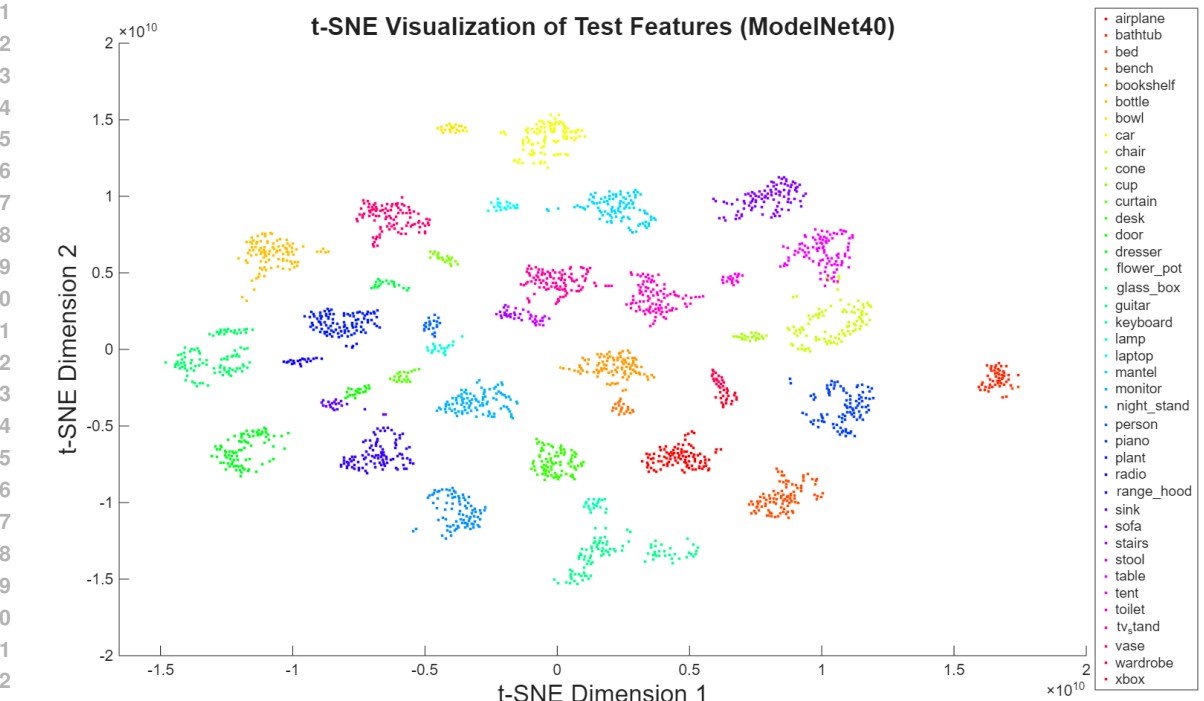

Figure 10: t-SNE analysis of ModelNet40 using the activations from the last RELU layer of our proposed network.

Table 5: Classification accuracy (%) results on OmniObject3D dataset Wu et al. (2023).

| Method | OA |
|---|---|
| DGCNN Wang et al. (2019) | 44.8 |
| PointNet Qi et al. (2017a) | 46.6 |
| PointNet++ Qi et al. (2017b) | 40.7 |
| RSCNN Liu et al. (2019c) | 39.3 |
| SimpleView Goyal et al. (2021) | 47.6 |
| GDANet Xu et al. (2021) | 49.7 |
| CurveNet Xiang et al. (2021) | 50.0 |
| PCT Guo et al. (2021) | 45.9 |
| RPC Ren et al. (2022) | 47.2 |
| TACO-Net (Ours) | **58.9** |

### 4.3.2 COMPARISON WITH NON-DEEP LEARNING ALGORITHMS

We show that the proposed 1D CNN model thoroughly outperforms standard non-deep learning methods such as XGBoost Chen & Guestrin (2016) and Random Forest classifiers. We chose the ModelNet40 dataset for this test while using the topological feature vector length of 1728 as mentioned in the paper. For XGBoost

Table 6: Comparison of OBJ_BG, OBJ_ONLY, and PB_T50_RS variants of the ScanObjectNN dataset against different *from-scratch supervised learning* methods.

| Methods | ScanObjectNN | | |
| --- | --- | --- | --- |
| | OBJ_BG | OBJ_ONLY | PB_T50_RS ↑ |
| PointNet Qi et al. (2017a) | 73.3 | 79.2 | 68.0 |
| PointNet++ Qi et al. (2017b) | 82.3 | 84.3 | 77.9 |
| DGCNN Wang et al. (2019) | 82.8 | 86.2 | 78.1 |
| PointCNN Li et al. (2018b) | 86.1 | 85.5 | 78.5 |
| MVTN Hamdi et al. (2021) | - | - | 82.8 |
| PointNeXt Qian et al. (2022) | - | - | 87.7 |
| PointMLP Ma et al. (2022) | - | - | 85.4 |
| ADS Hong et al. (2023) | - | - | 87.5 |
| X-3D Sun et al. (2024) | - | - | 90.7 |
| Mamba3D Han et al. (2024) | 94.49 | 92.43 | **92.64** |
| TACO-Net(Ours) | **95.08** | **94.01** | 91.87 |

Table 7: Comparison of our proposed TACO-Net and XGBoost on the six popular variants of the ScanObjectNN dataset Uy et al. (2019).

| Methods | OBG_ONLY | OBJ_BG | PB_T25 | PB_T25_R | PB_T50_R | PB_T50_RS |
| --- | --- | --- | --- | --- | --- | --- |
| XGBoost | 68.85 | 66.95 | 64.17 | 61.74 | 58.16 | 57.84 |
| TACO-Net(Ours) | 94.01 | 95.08 | 93.39 | 93.69 | 91.60 | 91.87 |

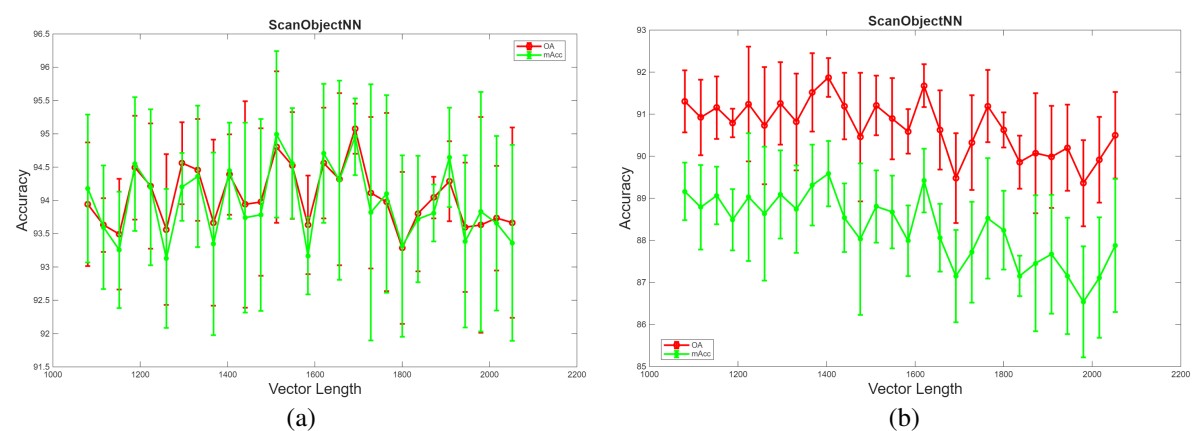

(a)  (b)

Figure 11: Change in OA/mAcc for the ScanObjectNN dataset w.r.t. feature vector length (i.e., different number of radial filtrations): a) OBJ_BG, b) PB_T50_RS.

and Random Forest classifiers, we used Python's xgboost and scikit-learn packages, respectively, with default options.

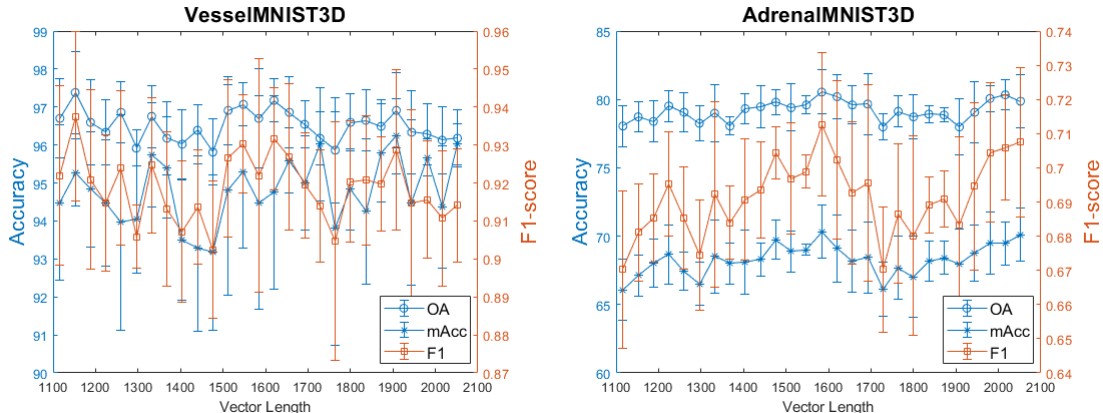

Figure 12: Change in performance with various radial center points and consequent feature vector lengths.

Table 8: VesselMNIST3D Yang et al. (2020): The best mAcc (%) and F1-scores from Yang et al. (2020) are reported along with our average results at the bottom.

| Network | mAcc | F1 |
|---|---|---|
| PointNet++ Qi et al. (2017b) | 93.52 | 0.90 |
| SpiderCNN Xu et al. (2018) | 92.59 | 0.87 |
| SO-Net Li et al. (2018a) | 91.50 | 0.88 |
| PointCNN Li et al. (2018b) | 92.38 | 0.90 |
| DGCNN Wang et al. (2019) | 90.67 | 0.86 |
| PointNet Qi et al. (2017a) | 81.62 | 0.69 |
| TACO-Net (Ours) | **95.28** | **0.94** |

Table 9: Overall accuracies (OA) in % for AdrenalM-NIST3D (A3D) and VesselMNIST3D (V3D) across different methods as benchmarked in Yang et al. (2023) compared with TACO-Net.

| Methods | A3D | V3D |
|---|---|---|
| ResNet-18 + 2.5D | 77.2 | 84.6 |
| ResNet-18 + 3D | 72.1 | 87.7 |
| ResNet-18 + ACS | 75.4 | 92.8 |
| ResNet-50 + 2.5D | 76.3 | 87.7 |
| ResNet-50 + 3D | 74.5 | 91.8 |
| ResNet-50 + ACS | 75.8 | 85.8 |
| auto-sklearn Feurer et al. (2015) | 80.2 | 91.5 |
| AutoKeras Jin et al. (2019) | 70.5 | 89.4 |
| TACO-Net (Ours) | **80.54** | **97.38** |

The comparison result is presented in Table 11. Our proposed TACO-Net achieves 17.7% and 20.7% higher accuracies than XGBoost and Random Forest, respectively. Notably, XGBoost required 74% more training time than that of TACO-Net. On the other hand, the throughput of XGBoost was 2.75x faster than TACO-Net. Overall, these results empirically demonstrate the superiority of the 1D CNN within the proposed TACO-Net framework in learning meaningful representations for object classification from the input topological vectors.

Table 10: Accuracies (%) when trained on clean ModelNet40 and tested on perturbed ModelNet40 test set.

| Perturbation | Low | | High | |
|---|---|---|---|---|
| | OA | mAcc | OA | mAcc |
| None (best model) | | – | 99.15 | 98.37 |
| Downsampling | 99.11 | 98.34 | 85.21 | 83.09 |
| Uniform | 98.82 | 97.17 | 70.10 | 63.08 |
| Gaussian | 94.08 | 90.35 | 54.50 | 47.58 |
| Upsampling | 91.90 | 88.02 | 50.28 | 47.19 |
| Rotation | 97.49 | 95.93 | 49.39 | 45.00 |
| Shear | 98.74 | 97.82 | 61.91 | 62.89 |
| FFD | 98.70 | 97.69 | 76.34 | 71.16 |
| RBF | 99.07 | 98.02 | 83.47 | 77.07 |
| Inverse-RBF | 99.15 | 98.47 | 86.63 | 80.04 |
| Impulse | 52.88 | 41.80 | 24.19 | 18.82 |

Table 11: Comparison with XGBoost and Random Forest classifiers (dataset: ModelNet40)

| Algorithm | OA (%) | Training Time (mins.) | Test Throughput |
|---|---|---|---|
| Random Forest | 78.35 | **0.55** | 18,985 |
| XGBoost | 81.35 | 4.35 | **61,700** |
| TACO-Net (Ours) | **99.05** | 2.50 | 16,454 |

Table 12: Comparison of network parameters (in millions) of different models for the ModelNet40 dataset.

| Methods | Param. (M) |
|---|---|
| PointNet Qi et al. (2017a) | 3.5 |
| PointNet++ Qi et al. (2017b) | 1.5 |
| MVTN Hamdi et al. (2021) | 3.5 |
| DGCNN Wang et al. (2019) | 1.8 |
| PointNeXt Qian et al. (2022) | 1.4 |
| PCT Guo et al. (2021) | 2.9 |
| Point-BERT Yu et al. (2022) | 22.1 |
| PointGPT Chen et al. (2023) | 29.2 |
| PointMamba Liang et al. (2024) | 12.3 |
| TACO-Net (Ours) | **0.72** |

Table 13: Shape retrieval (mAP) results on ModelNet40.

| Method | Retrieval mAP |
|---|---|
| 3D ShapeNets Wu et al. (2015) | 49.2 |
| Densepoint Liu et al. (2019b) | 88.5 |
| PVNet You et al. (2018) | 89.5 |
| MVCNN Su et al. (2015) | 80.2 |
| MLVCNN Yu et al. (2018) | 92.2 |
| MVTN Hamdi et al. (2021) | 92.9 |
| Latformer He et al. (2024) | 97.4 |
| TACO-Net (ours) | **99.33** |