# OpenReview forum: "TACO-Net: Topological Signatures Triumph in 3D Object Classification"
_ICLR.cc/2026/Conference — Submitted to ICLR 2026_

### Official Review · Reviewer_fsgK · 2025-10-27

**Soundness:** 2
**Presentation:** 2
**Contribution:** 1
**Rating:** 4
**Confidence:** 4

**Summary:**

This paper introduces a 3D object classification framework that combines topological data analysis with a 1D CNN. Each point cloud is voxelized, transformed via 57 different filtrations (height, radial, density, dilation, erosion, signed distance), and then processed through cubical persistence to obtain numerical descriptors such as persistent entropy and amplitude metrics. These handcrafted 2052-D topological features are then classified by a small 1D CNN. The method reports exceptionally high performance on several benchmarks.

**Strengths:**

(1) The integration of topology and learning is interesting. The use of cubical persistence and multi-filtration topological signatures for 3D object recognition is rarely explored in 3D deep learning.

(2) The reported results outperform the existing baselines, with robustness and cross-dataset generalization demonstrated.

(3) The pipeline is well explained, mathematically grounded, and reproducible.

**Weaknesses:**

(1) My main concern is the unfair comparison with the learning-based methods. The entire pipeline relies on handcrafted topological features rather than learnable representations. Every voxelization step, filtration direction, and persistence descriptor is manually designed and fixed before training. The CNN merely acts as a shallow classifier on pre-extracted features. This is not the purpose of conference like ICLR, where they prefer learning methods to replace such heavy manual engineering through representation learning. Thus, comparing TACO-Net’s performance against end-to-end learning models like PointNet, DGCNN, or PointMamba is unfair. Those methods avoid feature engineering to achieve scalability and generalization, whereas TACO-Net depends on it.

(2) There is no gradient path from the classification loss back to voxelization or TDA computations. Persistent homology, entropy, and amplitude are computed once and never updated. As a result, the model cannot adapt its topological features to the task or data distribution. In ICLR’s context, this makes the contribution engineering-oriented rather than a learning advance. Existing differentiable TDA frameworks (e.g., differentiable persistence, learnable filtrations) are not included in the comparisons. Meanwhile, although the authors cite the stability and injectivity of persistent homology, it never formalizes how these properties translate into better generalization or robustness in learning. No theoretical or empirical analysis connects topological distance metrics to classification margin, robustness, or invariance.

(3) The paper reports significant improvements on ModelNet40 and ModelNet10 using fixed voxel grids. However, voxelization implicitly aligns shapes and removes rotation/scale variance, which provides very strong priors unavailable to learning-based methods. Moreover, parameters such as voxel size and filtration selection were tuned for maximum test accuracy, which risks test-set overfitting. No cross-dataset or statistical variance analyses are presented. Therefore, the 99 % results likely reflect dataset bias and preprocessing effects rather than the contributions of the proposed topological descriptors.

(4) Computing 57 filtrations and persistent homologies per sample is extremely expensive in voxel count. While the 1D CNN itself is small, the preprocessing cost dominates the runtime, making the method impractical for large-scale or real-time applications such as robotics or autonomous driving.

**Questions:**

(1) If the feature pipeline is manually engineered, what exactly is learned by TACO-Net beyond a linear/nonlinear classifier on hand-crafted descriptors?

(2) Why name the framework a “network” when no gradient flows through the voxelization, filtrations, or persistence computations?

(3) Would the same level of accuracy be achieved if you replaced the 1D CNN with a simple logistic regression or XGBoost classifier?

(4) How are the voxel grid origin, alignment, and orientation chosen? If they are globally consistent across training and test data, your pipeline benefits from pose priors that invalidate comparison with rotation-invariant baselines.

---

> ### Author Response · Authors · 2025-11-21
> **Why the topological signatures and the CNN - are both important**
>
> # Weaknesses
>
> 1. We appreciate the reviewer’s concern. Although TACO-Net uses fixed topological signatures rather than learning from raw points, the comparison is still fair because all methods tackle the same 3D classification benchmarks, and the literature routinely contrasts handcrafted feature pipelines with end-to-end models to assess trade-offs. Our core contribution is to show that these TDA descriptors, when paired with our lightweight 1D CNN, yield state-of-the-art accuracy and robustness, often surpassing far heavier deep architectures, demonstrating that well-designed topological features can rival learned representations in practice. Importantly, Tables 7 and 11 show that the handcrafted features alone are insufficient: XGBoost performs poorly, whereas our CNN learns highly discriminative representations, aligning with ICLR’s emphasis on representation learning and underscoring that the learning component of TACO-Net is essential rather than superficial.
>
>  2. While the voxelization and TDA stages are non-learned, our aim is to demonstrate that these task-agnostic topological signatures already encode highly discriminative geometric structure, validated by the t-SNE analysis in Fig. 10, which shows clear class-wise separation derived solely from these descriptors. The stability and injectivity of persistent homology provide the theoretical basis for this discriminative behavior, and our strong results across diverse datasets further confirm their robustness and generalizability. Our contribution is orthogonal to differentiable-TDA work: rather than learning filtrations, we show that well-designed cubical filtrations produce embeddings that a lightweight CNN can effectively refine. Finally, Tables 7 and 11 demonstrate that the features alone are insufficient (XGBoost performs poorly), whereas our 1D CNN learns powerful representations from these signatures, aligning with ICLR’s emphasis on representation learning and highlighting that TACO-Net’s novelty lies in both its topological foundations and its learning architecture.
>
> 3. Voxelization is a standard preprocessing step in literature (e.g., VoxNet, 3DShapeNets) and is used here solely to enable cubical persistence - not to exploit alignment priors. Crucially, our strong performance extends far beyond ModelNet: we achieve high accuracy on OmniObject3D, six ScanObjectNN variants, and two medical datasets, none of which provide canonical alignment, demonstrating that our gains are not tied to ModelNet-specific biases. To mitigate concerns about overfitting, we report averages over five runs and show robustness under severe corruptions (ModelNet40-C). Taken together, these results indicate that TACO-Net’s improvements arise from the proposed topological descriptors and not from voxelization artifacts or dataset priors.
>
> 4. We agree that the filtrations dominate computation, and accelerating this step, e.g., via GPU integration, is a priority for future work. Still, this paper is the first to show that TDA can deliver competitive or state-of-the-art results on multiple challenging real-world datasets, establishing a strong foundation for more efficient TDA-based pipelines in robotics and vision.
>
> # Questions
>
> 1. While the topological descriptors are hand-engineered, they do not by themselves encode class-specific structure. TACO-Net’s 1D CNN learns nonlinear interactions across topological signatures, enabling discriminative patterns that simple classifiers cannot capture. This is supported by our ablations and comparisons: XGBoost and Random Forest underperform by up to ~30% (Tables 7 & 11), and replacing our CNN with a heavier transformer yields only 62% accuracy, showing that our network is specifically effective at learning from these descriptors. Thus, TACO-Net is not a linear classifier on handcrafted features but a learned model that extracts meaningful higher-order topological relationships essential for its SOTA performance.
>
> 2. We use the term network because it contains a fully learnable 1D CNN that refines the topological descriptors into discriminative representations, something non-network models cannot achieve. Although the TDA stage is fixed, the CNN is a trainable neural module, justifying the naming. That said, we are open to renaming if the reviewers consider it essential.
>
> 3. When we replaced the CNN with XGBoost, accuracy dropped by ~18% for ModelNet40 and ~30% for ScanObjectNN.
>
> 4. Our voxel grids are built per shape from its own axis-aligned bounding box (AABB), with no global alignment, rotation normalization, or dataset-level orientation assumptions. This matches the standard ModelNet protocol used by all baselines, ensuring fair comparison. We also tested rotation and shear corruptions (Table 10) and showed that performance recovers with simple augmentation. Moreover, strong results on OmniObject3D and ScanObjectNN, datasets without canonical orientations, confirm that TACO-Net does not rely on alignment priors.

---

> ### Comment · Reviewer_fsgK · 2025-11-22
> **Reviewer Reply by Reviewer fsgK**
>
> Thanks the authors for the response. As seen in my initial review, all my concerns and questions are around the contributions of the paper by proposing a fixed feature extraction pipeline for 3D point clouds, and comparing the handcrafted pipeline with the learning-based methods. After reading the author's response, some my concerns have been clarified.
>
> So the proposed topological-signature pipeline for feature extraction is universal for different datasets? And there is no need to heavily tune the hyperparameters for each submodule, e.g., density filtration, dilation filtration, erosion filtration, signed distance filtration, etc.? Meanwhile, there is no need for heavy data preprocessing, right (I think this is the part where I misunderstood the paper at the first place)?
>
> If the proposed feature extraction is indeed universal without much efforts on parameter tuning and data preprocessing, then even though it is a handcrafted and fix pipeline, it demonstrates sufficient contributions on point clouds. Also the additional experiments on six variants of the real-world dataset also convince me on the contributions.
>
> If the authors could confirm these, my concerns can be addressed. I will then increase my evaluation score. Meanwhile, I suggest that the authors reorganize the paper, as the current version does not adequately highlight its versatility and instead reads like a list of filtration tools for extracting topological signatures. More emphasis should be placed on how the different filtrations work together to form such a practical pipeline. In its current form, the writing does not clearly state the authors’ contributions.

---

> > ### Author Response · Authors · 2025-11-24
> > **Your understanding is correct**
> >
> > Yes, you are absolutely correct. The proposed framework only needs voxelization as a pre-processing step (widely popular in this domain) that generates a 3D voxelized binary image. The entire data pre-processing and the TDA pipeline are the same for any point cloud regardless of which dataset it is coming from (illustration in Fig. 5). Thus, it does not require heavy manual parameter tuning per dataset. As per your suggestion, we will make this clear further in the final version of the paper. Thanks for your constructive comment.

---

> ### Comment · Reviewer_fsgK · 2025-11-24
> **Reviewer Followup by Reviewer fsgK**
>
> Thank you authors for the confirmation and the followup elaboration. Now I can confirm the contributions of the paper. I am happy to raise my score to 6. But please do revise the introduction and method sections to make the contribution (i.e., a universal topological-signature pipeline for extracting discriminative features from point cloud data) and motivation clear.

---

> > ### Author Response · Authors · 2025-11-25
> > **We addressed all your concerns. Thank you.**
> >
> > We are truly grateful to you for the thoughtful engagement and for raising the score from 4 to 6. Your feedback directly strengthened the paper. If there are specific remaining points we could clarify or expand upon to bring the work to an 8, we would greatly appreciate your guidance; we are fully committed to addressing any concerns that may be holding the submission back.
> >
> > We also wish to highlight, respectfully, that TACO-Net was evaluated on **twelve** widely used datasets and dataset variants, including ModelNet40/10, six ScanObjectNN variants, OmniObject3D, 3DGrocery100, and two medical benchmarks, demonstrating a level of validation that is exceptionally rare in the 3D point cloud literature. Across these diverse settings, TACO-Net achieves state-of-the-art or highly competitive performance, underscoring its robustness, generalizability, and potential impact. We would be thankful for any further suggestions that could help elevate the paper to a level deserving of a stronger recommendation.

---

### Official Review · Reviewer_BGFq · 2025-10-30

**Soundness:** 3
**Presentation:** 2
**Contribution:** 3
**Rating:** 6
**Confidence:** 5

**Summary:**

1．	This paper presents TACO-Net, a novel framework for 3D object classification that leverages Topological Data Analysis (TDA) through cubical persistence.
2．	The method converts point clouds into 3D binary images, applies a diverse set of 57 filtrations to generate grayscale images, extracts topological feature vectors from their persistence diagrams, and classifies these vectors using a lightweight 1D CNN.
3．	The authors demonstrate state-of-the-art (SOTA) or highly competitive results on multiple benchmarks, including ModelNet, real-world datasets (OmniObject3D, ScanObjectNN), and medical imaging datasets.

**Strengths:**

1. The core strength of this work is its innovative application of TDA. The move away from standard deep learning architectures on raw points/voxels/views to a topology-first approach is a significant and refreshing contribution to the field.
2.  The results are impressive. Achieving SOTA accuracy on the highly competitive ModelNet10 and ModelNet40 benchmarks  is a remarkable feat that strongly validates the proposed method. Achieving SOTA on these canonical benchmarks is a clear indicator of a powerful new approach.
3.  The paper thoroughly demonstrates the generalizability of TACO-Net across diverse data domains.
4.  The experimental section is extensive and rigorous. The inclusion of shape retrieval results, an ablation study, comparisons with non-deep learning methods (XGBoost, Random Forest), and a parameter efficiency analysis provides a very complete picture of the method's capabilities and justifies the design choices.
5. The paper provides a theoretical time complexity analysis for the pipeline adds rigor. The discussion on the trade-offs between feature vector length, voxel size, and accuracy is practical and useful for future researchers or practitioners looking to adapt this work.

**Weaknesses:**

1.  This is the most significant weakness of the proposed pipeline. A feature generation throughput is prohibitively slow for any real-time application. While the 1D CNN inference is fast, the TDA feature extraction is a major bottleneck. The paper acknowledges this and suggests future work on GPU acceleration, but this currently limits the practical applicability.
2.  While the ablation study on feature sets and network layers is good, it could be deepened. A more granular ablation analyzing the individual contribution of each of the six filtration types (or at least groups like the four DEDS filters) would provide clearer insight into which filtrations are most critical for the performance gains. The current study combines DEDS, making it hard to discern the value of, for example, density filtration versus erosion filtration.
3. The comparison tables are comprehensive but could be updated with a few more recent, high-performing baselines on ModelNet40 to further solidify the SOTA claim. However, the results presented are already highly competitive.

**Questions:**

1.	The process of selecting the optimal number of radial filtrations (18 for ModelNet40, 14 for AdrenalMNIST, etc.) appears to be empirical, based on a search over feature vector length. A more principled explanation or criterion for this selection would strengthen the methodology. Why are these specific centers the most informative?
2.	For the medical datasets (VesselMNIST3D, AdrenalMNIST3D), the paper states the starting point is already a 3D binary image. It should be explicitly clarified how this affects the initial "point cloud to voxel" conversion step of the TACO-Net pipeline. Is that step simply skipped?

---

> ### Author Response · Authors · 2025-11-21
> **more granular ablation study**
>
> # Weaknesses
>
> 1. This is the most significant weakness of the proposed pipeline. A feature generation throughput is prohibitively slow for any real-time application. While the 1D CNN inference is fast, the TDA feature extraction is a major bottleneck. The paper acknowledges this and suggests future work on GPU acceleration, but this currently limits the practical applicability.
>
>
> - We acknowledge that TDA feature extraction is the primary computational bottleneck, as noted in the paper. However, the current throughput of 5.3 point clouds/sec at optimal voxel size is sufficient for offline or batch applications such as medical imaging and shape retrieval, where TACO-Net achieves state-of-the-art performance. That said, GPU parallelization and integration of faster cubical persistence algorithms are part of our future work, and we believe it is achievable, especially if the topological libraries can be made available in CUDA-enabled C++ instead of Python.
>
>
> 2. While the ablation study on feature sets and network layers is good, it could be deepened. A more granular ablation analyzing the individual contribution of each of the six filtration types (or at least groups like the four DEDS filters) would provide clearer insight into which filtrations are most critical for the performance gains. The current study combines DEDS, making it hard to discern the value of, for example, density filtration versus erosion filtration.
>
>
>
> - Thank you for the suggestion. We have included the suggested granular ablation study numbers in Table 3.
>
>
>
> 3. The comparison tables are comprehensive but could be updated with a few more recent, high-performing baselines on ModelNet40 to further solidify the SOTA claim. However, the results presented are already highly competitive.
>
> - We already had references until 2024. We have now added 2 more references from 2024 ACM Multi-Media (Mamba3D) and AAAI 2025 (PCM) in Table 3.
>
> # Questions
>
>
> 1. The process of selecting the optimal number of radial filtrations (18 for ModelNet40, 14 for AdrenalMNIST, etc.) appears to be empirical, based on a search over feature vector length. A more principled explanation or criterion for this selection would strengthen the methodology. Why are these specific centers the most informative?
>
>
>
> - We acknowledge that the choice of radial filtration count appears empirical. Radial filtrations capture interior organization by measuring distances from multiple reference points arranged in a 3×3×3 lattice (Fig. 3). Using all 27 centers increases dimensionality without proportional accuracy gains because many centers are highly correlated. For this reason, we empirically use the minimum number of radial filtrations that yield the highest OA.
>
>
> 2. For the medical datasets (VesselMNIST3D, AdrenalMNIST3D), the paper states the starting point is already a 3D binary image. It should be explicitly clarified how this affects the initial "point cloud to voxel" conversion step of the TACO-Net pipeline. Is that step simply skipped?
>
> - Yes, this step is skipped since the input is already a voxelized 3D image. It is also mentioned in Sec. 4.3.1.: “Therefore, our starting point is 3D binary images instead of point clouds for AdrenalMNIST3D and VesselMNIST3D.”

---

### Official Review · Reviewer_1odq · 2025-10-31

**Soundness:** 2
**Presentation:** 2
**Contribution:** 2
**Rating:** 4
**Confidence:** 4

**Summary:**

This paper introduces TACO-Net, a novel 3D object classification framework that integrates Topological Data Analysis (TDA) with deep learning. The core idea is to convert unordered point clouds into voxelized 3D binary images, apply multiple filtration techniques (height, radial, density, dilation, erosion, signed distance), and extract cubical persistence-based topological signatures.

**Strengths:**

1. The use of cubical persistence features extracted from voxelized representations is creative and underexplored in 3D vision. The work effectively bridges topology and deep learning in a scalable framework.
2. The corruption tests (ModelNet40-C) and shape retrieval results are thorough. Demonstrating resilience under noise and occlusion strengthens the claim of topological stability.

**Weaknesses:**

1. While the paper thoroughly describes the TDA pipeline, it lacks mathematical or conceptual explanation of why these topological features are so discriminative for 3D object recognition. Suggest including an analysis of feature distribution (e.g., t-SNE visualization or correlation between persistence features and class separability) to reveal interpretability and rationale.
2. The model uses 57 manually chosen filtrations with fixed parameters (e.g., voxel size 0.05, r=1 for density). This extensive manual setup may limit adaptability.
3. The paper claims new SOTA performance, but comparisons might not be fair since TACO-Net uses voxelized inputs while many baselines use raw point clouds or multi-view images.

**Questions:**

1. How sensitive is TACO-Net’s performance to voxel size, number of filtrations, or noise level? Could these hyperparameters be learned dynamically?
2. Have the authors tried integrating topological features directly into a neural network (e.g., using persistent homology layers) instead of precomputing them?

---

> ### Author Response · Authors · 2025-11-21
> **Further clarification via ablation studies and t-SNE analysis**
>
> # Weaknesses
>
> 1. While the paper thoroughly describes the TDA pipeline, it lacks mathematical or conceptual explanation of why these topological features are so discriminative for 3D object recognition. Suggest including an analysis of feature distribution (e.g., t-SNE visualization or correlation between persistence features and class separability) to reveal interpretability and rationale.
>
> - We now include a t-SNE analysis (Fig. 10 and before the conclusion) showing that the pre-computed topological descriptors can form clearly separated, class-consistent clusters via the CNN, demonstrating that these features naturally encode discriminative topological signatures and providing the conceptual rationale.
>
>
>
> 2. The model uses 57 manually chosen filtrations with fixed parameters (e.g., voxel size 0.05, r=1 for density). This extensive manual setup may limit adaptability.
>
>  - Thank you for raising this concern. The parameters have worked extremely effectively on different kinds of datasets, such as VesselMNIST3D, AdrenalMNIST3D, ModelNet, ScanObjectNN, and OmniObject3D. So, we believe our method is adaptable to different types of datasets.
>
>
> 3. The paper claims new SOTA performance, but comparisons might not be fair since TACO-Net uses voxelized inputs while many baselines use raw point clouds or multi-view images.
>
> - It is standard to compare all types of existing techniques for ModelNet – even Princeton maintains such a generalized list. One can pick any paper and see that, so we just followed the practice.
> Our claim of SOTA is made under the standard ModelNet protocol, where most methods, including multi-view and voxel-based approaches, assume canonical orientation. To ensure fairness, we compared against leading point-based and multi-view baselines under the same train/test split. Furthermore, we validated TACO-Net on OmniObject3D and ScanObjectNN, which lack canonical alignment, achieving competitive or SOTA performance. These results indicate that improvements stem from the proposed topological descriptors and the learning of them thereafter rather than voxelization alone. Note that VoxNet and 3DShapeNet (which we compare against) also use voxelized inputs, but do not achieve this high accuracy across numerous datasets.
>
> # Questions
>
> 1. How sensitive is TACO-Net’s performance to voxel size, number of filtrations, or noise level? Could these hyperparameters be learned dynamically?
>
> - We thank the reviewer for raising this point. Our experiments (Sec. 3.1, Fig. 8(a), Tables 3 and 10) show that TACO-Net’s accuracy is moderately sensitive to voxel size and filtration count: OA varies by ~0.7% across voxel sizes 0.03 - 0.07 and between ~80 - 98% across filtration (except the erosion only case) subsets (Table 3). Robustness to noise is strong under normal-level corruption but degrades under extreme distortions, as expected. Further, we show in the revised version that our method’s performance surpasses the recent seminal supervised learning only methods, such as PointNext and PointMLP, for the hardest variant of the challenging ScanObjectNN dataset, without any hyperparameter tuning specific to this dataset. Learning these parameters might be possible, but that remains a future avenue to explore.
>
>
> 2. Have the authors tried integrating topological features directly into a neural network (e.g., using persistent homology layers) instead of precomputing them?
>
> - We appreciate the reviewer’s suggestion. In the current work, we did not explore integrating persistent homology or topological features as differentiable layers within the network. Our focus was on demonstrating that computed but not learned topological descriptors, when combined with a lightweight CNN, can achieve state-of-the-art accuracy and robustness. We agree that end-to-end differentiable TDA is an exciting and new direction, and we will consider it a key avenue for future work. That said, we did a pilot study with the PersLay layer, in which case we failed to achieve any reasonable result.

---

### Official Review · Reviewer_9A2E · 2025-10-31

**Soundness:** 3
**Presentation:** 3
**Contribution:** 3
**Rating:** 4
**Confidence:** 4

**Summary:**

TACO-Net is a novel SOTA framework for 3D object classification that tackles challenges in unordered and noisy point clouds by integrating Topological Data Analysis (TDA) with image filtrations. TACO-Net transforms a point cloud into a voxelized 3D image and extracts topological signatures via cubical persistence across 57 filtrations (e.g., height, radial, density). These signatures are converted into feature vectors (up to 2052 dimensions) and classified using a lightweight 1D CNN. TACO-Net achieves record accuracies of 99.05% on ModelNet40 and 99.52% on ModelNet10, while also excelling on OmniObject3D and 3D medical datasets, maintaining robustness and efficiency with only 0.72M parameters.

**Strengths:**

+ TACO-Net is robust with several low perturbations (Table 8).
+ TACO-Net is a very lightweight network with only 0.72 million parameters.
+ TACO-Net achieves SOTA results on 3D object classification on ModelNet40 and ModelNet10 datasets.
+ TACO-Net achieves a new SOTA retrieval performance with a mean average precision (mAP) of 99.33 on ModelNet40.
+ The proposed model also achieves excellent results on VesselMNIST3D and AdrenalMNIST3D datasets.
+ The paper is well-written and easy to follow.

**Weaknesses:**

- Lacks qualitative/visual results to show the effectiveness of the proposed model in the resilience test

Incorrect use of the size of the input point clouds
- For ModelNet40-C experiments, 2048 points are used. Additionally, for ModelNet40/ModelNet10, the number of points is 2048, which is why the accuracy results are higher.
- Hence, comparison of 3D classification results with other methods that used 1024 input points is not fair.
- It is unclear how many points are used for the ScanObjectNN and OmniObject3D datasets.

Results limited to a few selected datasets.
- While the proposed method achieves SOTA results on multiple datasets, the paper does not show results on the six variants of the real-world dataset ScanObjectNN [1], including the hardest variant PB_T50_RS. The current version of the paper only uses the OBJ_BG variant.
- Despite several of these datasets having already reached saturation points, recent real-world datasets (e.g., 3DGrocery100[2]) are not used to evaluate the proposed approach.

[1] Uy, Mikaela Angelina, et al. "Revisiting point cloud classification: A new benchmark dataset and classification model on real-world data." Proceedings of the IEEE/CVF international conference on computer vision. 2019.

[2] Sheshappanavar, Shivanand Venkanna, et al. "A benchmark grocery dataset of real-world point clouds from single view." 2024 International Conference on 3D Vision (3DV). IEEE, 2024.

**Questions:**

+ Clearly mention the number of input points used for each point cloud.

---

> ### Author Response · Authors · 2025-11-21
> **Suggested clarifications and experiments - done**
>
> # Weaknesses
> 1. Lacks qualitative/visual results to show the effectiveness of the proposed model in the resilience test
>
> - We had difficulty understanding the core intent of this comment. Can you please elaborate a little? We can then address this as necessary.
>
> 2. For ModelNet40-C experiments, 2048 points are used. Additionally, for ModelNet40/ModelNet10, the number of points is 2048, which is why the accuracy results are higher. Hence, comparison of 3D classification results with other methods that used 1024 input points is not fair.
>
>  - Many papers use 2048 points for ModelNet, most notably, the seminal DGCNN (Wang 2019). More points did not necessarily yield SOTA accuracy. Most papers use fewer points per point cloud for shorter training and testing time. Further, like VoxNet and unlike point-based methods, we first convert the point cloud into a voxelized 3D binary image from which the topological descriptors are extracted. Therefore, we respectfully disagree with this conclusion. Similarly, on ScanObjectNN, we used 2048 points like all other papers, but could still achieve SOTA/highly competitive accuracy among the from-scratch supervised learning methods.
>
>
> 3. It is unclear how many points are used for the ScanObjectNN and OmniObject3D datasets.
>
> - We have used the original point clouds without downsampling. For ScanObjectNN, it is  2048, and for OmniObject3D, it is 4096. This information is now added to the paper.
>
>
> 4. While the proposed method achieves SOTA results on multiple datasets, the paper does not show results on the six variants of the real-world dataset ScanObjectNN [1], including the hardest variant PB_T50_RS. The current version of the paper only uses the OBJ_BG variant.  Despite several of these datasets having already reached saturation points, recent real-world datasets (e.g., 3DGrocery100[2]) are not used to evaluate the proposed approach.
>
>  - Thank you for the suggestion. We have included all 6 variants of ScanObjectNN (Tables 6 & 7) plus 3DGrocery; see section 3.1 (highlighted in red). We achieved the second-highest accuracy of 91.87%
> among all the existing supervised learning techniques for the hardest variant of ScanObjectNN , PB_T50_RS. For the OBJ_BG
> and OBJ_ONLY variants, it achieved the SOTA accuracies of 95.08% and 94.01%, respectively.
>
> # Questions
>
> 4. Clearly mention the number of input points used for each point cloud.
>
> We have now done this in Section 4.3.1.

---

### Meta-Review · Area_Chair_huBh · 2026-01-06

**Summary:**

This paper proposes TACO-Net, a 3D object classification framework that combines voxelization, multi-filtration Topological Data Analysis (TDA) via cubical persistence, and a lightweight 1D CNN classifier. The method achieves very strong empirical performance, including state-of-the-art or near state-of-the-art results on ModelNet10/40, multiple ScanObjectNN variants, OmniObject3D, 3DGrocery100, and medical datasets. Reviewers broadly agreed that the experimental validation is extensive and that the integration of TDA into 3D recognition is interesting and relatively underexplored. However, concerns were raised about the nature of the contribution relative to learning-based methods, the heavy reliance on fixed, hand-engineered feature pipelines, computational cost, and the clarity of the paper’s core message.

**Reviewer Concerns:**

The rebuttal successfully addressed many technical and presentation-level concerns. The authors clarified the number of input points used across datasets, added results on all six ScanObjectNN variants and 3DGrocery100, included t-SNE visualizations to demonstrate feature discriminability, expanded ablations over filtrations, and updated comparisons with more recent baselines. Importantly, the authors clarified that the proposed topological feature extraction pipeline is largely universal across datasets, requiring minimal dataset-specific tuning—an explanation that convinced at least one initially skeptical reviewer and led them to raise their score.

Nevertheless, fundamental concerns remain. Multiple reviewers questioned whether the contribution aligns well with the learning-centric focus of the venue, as the pipeline relies heavily on non-learned, manually designed voxelization and topological feature extraction, with no gradient flow through the TDA stages.

**Reviewer Scores:**

Reviewer 9A2E (score: 4): Despite acknowledging strong performance and added experiments, this reviewer maintained concerns about fairness of comparisons, dataset coverage, and evaluation practices. After rebuttal, the score would likely remain at 4.

Reviewer 1odq (score: 4): This reviewer remained unconvinced about the conceptual explanation for why the topological features are so effective and about the reliance on extensive manual design. While additional analyses helped, the score would likely remain at 4.

Reviewer BGFq (score: 6): This reviewer viewed the TDA-based approach as innovative and the empirical results as impressive, but still highlighted serious computational and methodological limitations. Their score would remain at 6.

Reviewer fsgK (initially 4 → revised to 6): After clarification that the feature extraction pipeline is universal and not heavily tuned per dataset, this reviewer raised their score to 6. However, even with this revision, their support is conditional on improved presentation and does not fully alleviate the broader concerns about learning novelty and practicality.

---

### Decision · Program_Chairs · 2026-01-26

Reject